# EXPOSING VULNERABILITIES IN LATENT-NOISE DIFFUSION WATERMARKS

## ABSTRACT

Watermarking techniques are crucial for protecting intellectual property and preventing the fraudulent use of media. Recently, a prominent approach to watermarking diffusion models relies on embedding a secret key in the initial noise. The resulting pattern is often considered hard to forge into unrelated images and remove. In this paper, we make a key observation that there is an inherent many-to-one mapping between images and initial noises. Therefore, there are regions in the clean image latent space pertaining to each watermark that get mapped to the same initial noise when inverted. We expose this as a vulnerability by proposing a black-box adversarial attack that uses only a single watermarked image without presuming access to any diffusion model. Our forgery attack simply adds perturbations to unrelated, potentially harmful images so that they would enter the region of watermarked images and get falsely labeled as watermarked. We show that a similar approach can also be applied to watermark removal by learning perturbations to exit from this region. We report results on multiple watermarking schemes (Tree-Ring, RingID, WIND, and Gaussian Shading). Our results demonstrate the effectiveness of the attack and expose vulnerabilities in current watermarking methods, motivating future research on improving them.

## 1 INTRODUCTION

Latent diffusion models have made significant strides in terms of producing realistic-looking images. However, this advancement comes with its own set of problems, primarily related to image provenance. Image forensics experts are tasked with verifying which generative model generated a particular image, if any, and who is responsible for any harm arising from it. Researchers and policymakers alike have focused on watermarking generative models as a potential solution to the threat from these models. This approach involves embedding an imperceptible pattern into images, allowing them to verify whether or not an image was generated using a specific model.

Image watermarking is a well-studied field (Potdar et al., 2005; Hartung and Kutter, 1999; Podilchuk and Delp, 2001); however, recent advances in generative modeling have transformed both watermarking techniques and attacks against them. To improve robustness against removal attacks, leading watermarking schemes in diffusion models have focused on embedding a secret key into the initial noisy latent vector (Wen et al., 2023; Ci et al., 2024b; Arabi et al., 2024; Yang et al., 2024b). This enables a model owner to generate a realistic image using the standard denoising process without altering the model weights or compromising image quality. During verification, the denoising diffusion implicit model (DDIM) inversion Mokady et al. (2023) is used to invert the image, recovering the initial noise sample and the initial pattern that may have been embedded. These methods have shown promising results in avoiding watermark removal. Yet, as we show in this paper, the process remains vulnerable to adversaries.

Attacks on watermarks fall into two main categories: *forgery attacks* (Yang et al., 2024a; Kaur et al., 2023; Wang et al., 2021) and *removal attacks* (Zhao et al., 2025; Lukas et al., 2023; Liu et al., 2024; Yang et al., 2024a; Hu et al., 2024; An et al., 2024). Forgery attacks attempt to steal the watermark and apply it to content unrelated to the model's owner, raising concerns about false attribution of harmful content. Removal attacks aim to remove the watermark while preserving the image content, raising concerns about intellectual property infringement or harmful use of the generated content to mislead the public. Previous attack methods have shown some success, but their

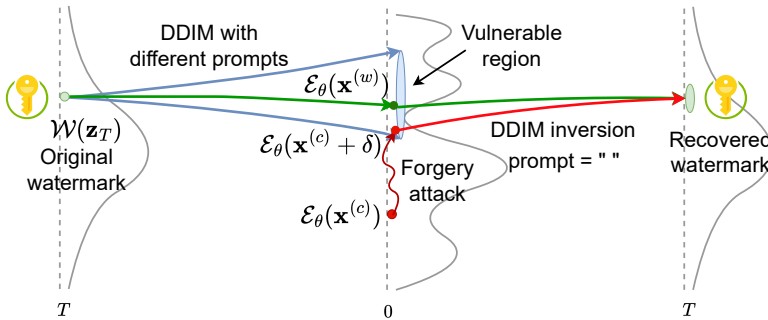

Figure 1: Intuition behind our attack: Within the latent space, an entire region maps to approximately the same key-embedded initial latent noise vector. An attacker only needs to ensure their sample embeds within this region to be falsely classified as watermarked.

success relied on one or more of the following conditions: (i) collecting a large set of watermarked images (and possibly, their non-watermarked counterparts), (ii) access to the entire model weights, or some approximation of it, and (iii) significantly distorting the attacked images.

In this paper, we start by making a key observation about the dependence on the DDIM noise inversion (Mokady et al., 2023). It is the fact that different prompts can utilize the same initial noise during generation, but the DDIM inversion process to recover the secret key takes place with an empty prompt. This fact implies that there is a many-to-one relationship from the clean denoised latent space to the initial noise latent space. This, in turn, suggests that there might be a non-trivial region in a Variational Auto-Encoder (VAE)'s latent space that corresponds to each watermark. We showcase this idea in Figure 1. We also verify that this hypothesis is true by using a linear support vector machine model to find latent directions corresponding to watermarked and non-watermarked images. Traversing along these directions allows easy forgery and removal of the watermark signal.

Building upon this, we propose an adversarial attack to forge a given watermark, wherein the attacker simply needs to find adversarial perturbations such that a target image gets embedded close enough to a watermarked example in the VAE's latent representation space. We show that an imperceptible modification to the image, which does not alter its semantic content, is sufficient due to the inherent non-smoothness of the VAE's representation space (Cemgil et al., 2020). Once this objective is achieved, the DDIM inversion process with an empty prompt will guide both of these latents to a similar initial noise state, successfully fooling the watermark detection system. We show a pictorial representation of our methodology in Figure 2. Moreover, we show that a similar method can be used for a removal attack, removing the watermark from an image while preserving the image content. In this scenario, our attack objective is to ensure that the latent representation for a watermarked image gets as close as possible to a non-watermarked image region, so that it leads to a false negative match when inverted.

Our method has several strengths, which make it easier for an adversary to attack the watermarking system. (1) Our method needs only one watermarked image to forge or remove a watermark. (2) Our method can achieve it without inverting an image to its initial noise state to tamper with the secret key, which would require access to a denoising network approximating the one used in generation. This distinction is key, as the denoising network (U-Net) is often fine-tuned for purposes such as filtering out not-safe-for-work or copyrighted content. (3) Although our method does require access to a VAE, it does not necessarily need access to the same VAE that was used by the watermarked diffusion model. A VAE that was trained on a similar dataset can suffice for this task.

To summarize, we make the following contributions:

- **Identifying Watermarked Latent Subspaces** – We show that there exist latent directions corresponding to each watermark pattern.
- **Watermark Forgery and Removal Attacks** – We propose attacks that rely *only* on a single watermarked image and a proxy VAE.
- **Impact Assessment** – We evaluate and minimize the impact of these attacks on the quality of the target image.

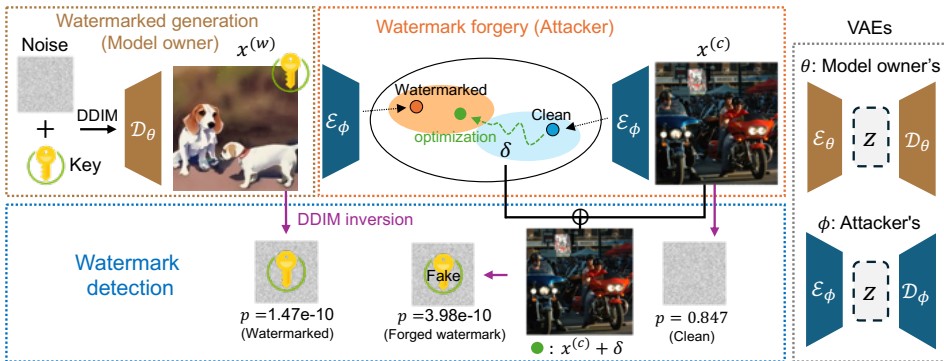

Figure 2: Our forgery attack works by finding an adversarial perturbation $\boldsymbol{\delta}$ such that the latent representation of the non-watermarked image $\mathcal{E}_\phi(\mathbf{x}^{(c)} + \boldsymbol{\delta})$ is close to the one corresponding to that of a watermarked image $\mathcal{E}_\phi(\mathbf{x}^{(w)})$. We do so while ensuring that we only introduce imperceptible changes to the clean image.

## 2 RELATED WORK

**Watermarking Schemes for Diffusion Models.** Watermarking is a technique that embeds a traceable mark in an image, enabling ownership verification and supporting media authentication and intellectual property protection.(Potdar et al., 2005; Hartung and Kutter, 1999; Podilchuk and Delp, 2001; Wong and Memon, 2001; Cox et al., 2007). Classical watermarking techniques have involved embedding an invisible pattern in an image that can be recovered. Similarly, in the context of generative models, watermarks can be embedded post hoc, after the image is generated (Wong and Memon, 2001; Tancik et al., 2020; Cox et al., 2007; Fernandez et al., 2022; Zhang et al., 2019). Alternatively, watermarking schemes in diffusion models have also focused on fine-tuning VAE decoders to watermark an output image (Ci et al., 2024a; Fernandez et al., 2023; Xiong et al., 2023).

A widely used approach to watermarking images involves embedding a secret key into the initial noise used by the diffusion model. Wen et al. (2023), who initially proposed this idea, showed that initial noise-based watermarking is more robust against removal attacks - transformation aiming to render the watermark undetectable (Zhao et al., 2025). Ci et al. (2024b) further improved the watermarking pattern structure to make it more secure. Yang et al. (2024b) utilized distribution-preserving sampling to ensure that the initial noise follows a Gaussian distribution, preserving the distribution of generated images. Arabi et al. (2024) showed that breaking keys into groups and using an initial pattern specific to the groups enables embedding a larger number of secret keys, improving security. Gunn et al. (2024) used a pseudo-random error correcting code to initialize the initial noise sample. These methods have focused on generating distortion-free images that come from a similar distribution as non-watermarked images. It was often implicitly assumed that such distortion-free watermarks would be more secure against various types of attacks. However, as we will show, this is not always the case.

**Forgery and Removal Attacks against Watermarks.** Forging and removing watermarking has been an important research area to expose vulnerabilities in watermarking schemes, and therefore lead to their improvement. Zhao et al. (2025) demonstrated that many watermarks can be removed by simply noising and then denoising a watermarked image using a diffusion model; however, their approach was unsuccessful against the Tree-Ring (Wen et al., 2023) watermarking method. Yang et al. (2024a) showed that methods such as Wen et al. (2023) leave distinct textural patterns in the image, which can be found by averaging multiple watermarked images. Müller et al. (2024) used an auxiliary diffusion model to ensure the inverted initial noises from a watermarked and non-watermarked image are closely aligned. WAVES (An et al., 2024) benchmarked different watermarking and attack methods to judge their effectiveness. They also proposed a set of attacks to evaluate the robustness of various watermarking schemes. Saberi et al. (2023) proposed training a proxy watermarked image classifier to classify whether an image is watermarked or not. They conducted a progressive gradient-descent-based adversarial attack on the model and showed that the perturbations were transferable to a black-box detection model. Liu et al. (2024) proposed re-

generating an image similar to watermarked images from clean Gaussian noise so as to remove the watermark. Lukas et al. (2023) proposed using differentiable surrogate keys to learn attack parameters, which enables the removal of traces of watermarked keys from images. This assumes access to not only a surrogate key generator but also a copy of the generative model.

In contrast to Saberi et al. (2023); Yang et al. (2024a), our method does not require access to multiple watermarked images. They assume access to images not only from the same watermarking method but also from the same secret key, making these attacks less practical against some systems (Arabi et al., 2024). We can run our attack using just one watermarked image. Furthermore, unlike Müller et al. (2024); Lukas et al. (2023), we do not assume any access to a denoising diffusion model or a proxy version of it. Lastly, in contrast to Zhao et al. (2025), which was unsuccessful in removing the Tree-Ring watermark, we demonstrate that our method achieves it effectively.

## 3 PRELIMINARIES

Diffusion models (Song et al., 2020) such as Stable Diffusion (SD) (Rombach et al., 2022) and Imagen (Saharia et al., 2022) learn a mapping from an initial random noise state $\mathbf{z}_T \sim \mathcal{N}(0, \mathbf{I})$ to a clean image space $\mathbf{z}_0 \sim p_{\text{data}}$. This is done by iteratively applying a learned denoising network $\boldsymbol{\epsilon}_\theta$ such as U-Net or DiT. Popularly used models (Rombach et al., 2022) compress the image space to a lower-dimensional representation space using a variational autoencoder (an encoder $\mathcal{E}_\theta$ and decoder $\mathcal{D}_\theta$) to reduce the amount of computation required for generating an image.

Using the learned noise estimator network $\boldsymbol{\epsilon}_\theta$, DDIM's sampling process (Song et al., 2020) computes the previous state $\mathbf{z}_{t-1}$ from $\mathbf{z}_t$ as follows:

$$\mathbf{z}_{t-1} = \sqrt{\frac{\bar{\alpha}_{t-1}}{\bar{\alpha}_t}} \mathbf{z}_t - \left( \sqrt{\frac{1}{\bar{\alpha}_{t-1}} - 1} - \sqrt{\frac{1}{\bar{\alpha}_t} - 1} \right) \boldsymbol{\epsilon}_\theta(\mathbf{z}_t, t, \boldsymbol{e}_{\text{p}}), \quad (1)$$

where $\beta_t$ is defined by the noise scheduler and $\bar{\alpha}_t = \prod_{i=1}^{t}(1 - \beta_i)$.

DDIM inversion (Mokady et al., 2023; Dhariwal and Nichol, 2021; Song et al., 2020) is a process to invert a clean sample $\mathbf{z}_0$ to reconstruct its initial noise state $\mathbf{z}_T$ based on the assumption that $\mathbf{z}_{t-1} - \mathbf{z}_t \approx \mathbf{z}_{t+1} - \mathbf{z}_t$. This allows us to estimate $\mathbf{z}_{t+1}$ from $\mathbf{z}_t$ using the formula,

$$\mathbf{z}_{t+1} = \sqrt{\bar{\alpha}_{t+1}} \mathbf{z}_0 + \sqrt{1 - \bar{\alpha}_{t+1}} \boldsymbol{\epsilon}_\theta(\mathbf{z}_t, t). \quad (2)$$

### 3.1 WATERMARKING SCHEME IN DIFFUSION MODELS

Most watermarking schemes in diffusion models consist of embedding a secret key $k \in \mathcal{K}$ in the initial noise $\mathbf{z}_T$ used to generate an image. This is done in a manner such that the initial noise does not deviate significantly from a standard Gaussian distribution $\mathcal{N}(0, \mathbf{I})$. The standard diffusion denoising process is followed to convert the given initial noise to a clean latent representation $\mathbf{z}_0$, which corresponds to a watermarked image $\mathbf{x}^{(w)}$ that will be perceptibly indistinguishable from non-watermarked images.

During detection, the DDIM inversion is used to estimate the initial noise sample $\mathbf{z}'_T$ from the clean sample $\mathbf{z}_0$. Once the initial noise is recovered, the key pattern (if any) is extracted and matched with the set of secret keys that a model owner used. It is important to note here that the inversion process is performed using an empty prompt, as the model owner typically does not keep track of the generated images or the prompts used. This process enables a model owner to watermark an image without requiring any modifications to the diffusion model architecture or weights. This approach has been shown to be robust against image transformations, which can significantly degrade other types of watermarking methods, specifically post-hoc watermarks (Wen et al., 2023).

## 4 WATERMARKING ATTACK TECHNIQUES

In this section, we introduce our watermarking attack techniques. We first define the threat model that we consider in Section 4.1, followed by explaining the motivation for our approach in Section 4.2. Lastly, we describe the adversarial attack itself in Section 4.3.

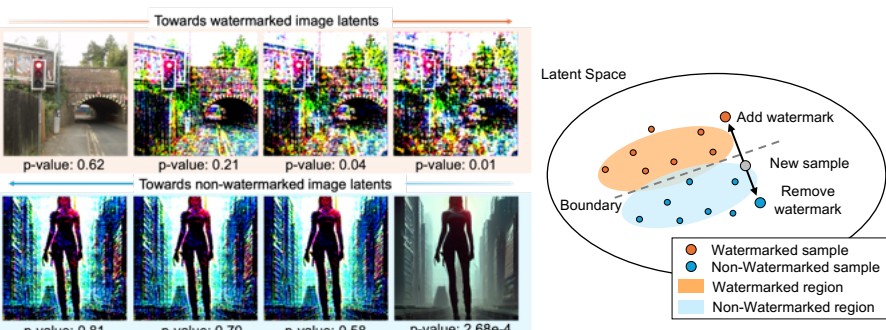

Figure 3: Motivation for our attack *(not explanation of our proposed attack itself)*. A latent direction exists pertaining to watermarked latents derived from a specific secret key in the clean image latent space. The further we traverse in the relevant direction, the stronger the attack becomes. Our method, proposed in Section 4.3, exploits this vulnerability. See Appendix Figure 13 for more examples.

## 4.1 THREAT MODEL

We consider two parties: a model owner and an attacker. The model owner owns a generative model and controls the generation such that it outputs watermarked images. The attacker is a party with malicious intentions that seeks to tamper with the watermarking system. One type of attacker may wish to *forge* the watermark into an unrelated image, to falsely claim that a harmful image was generated by the model owner. Another type of attacker may attempt to *remove* the watermark pattern from a previously watermarked image. This could be done to falsely claim ownership or to spread misinformation by concealing the image's synthetic origin, making it harder to detect it as AI-generated content (e.g., to create deepfakes). Formally:

- Model Owner (Generation Phase): The model owner owns a diffusion model $\epsilon_\theta$ and uses a random secret key $k$ from a set of secret keys $\mathcal{K}$ to generate a latent noise-based watermarked image $\mathbf{x}^{(w)}$.

- Attacker: The attacker wishes to use a single watermarked image $\mathbf{x}^{(w)}$ to falsely watermark a clean image $\mathbf{x}^{(c)}$ or to remove the watermark in $\mathbf{x}^{(w)}$ while preserving their contents. The attacker does not have access to the model $\epsilon_\theta$ used by the model owner or the secret key $k$ embedded by the model owner. We assume the watermarking method is a latent-noise-based one, and the attacker has access to a proxy VAE that was trained on a similar dataset. The attacker can get access to a watermarked sample by sampling the model, but every additional sampling is assumed to lead to an image with a different watermark key.

- Model Owner (Detection Phase): The model owner is asked to verify whether or not an image provided by the attacker was generated using their diffusion model $\epsilon_\theta$ by matching the extracted key to their set of secret keys $\mathcal{K}$.

## 4.2 MOTIVATION

Our approach is based on the intuition that the mapping from generated images to initial noise is inherently many-to-one, as the same initial noise sample can produce many different images when denoised using different prompts. In the case of watermark detection, the DDIM inversion process is performed using an empty prompt, meaning that all images generated with a specific secret key, regardless of the original text prompt, are expected to be inverted to recover that key. Based on this, we hypothesize that within the clean sample latent space, there exists a region that consistently maps to an initial noise pattern corresponding to the key. Our method focuses on exploiting this vulnerability: i.e., if we can successfully embed our non-watermarked sample in the watermarked region, we will be able to falsely claim that a non-watermarked

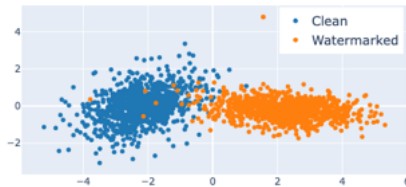

Figure 4: Two-dimensional visualization of the latent space showing the linear separability of watermarked and non-watermarked images. The horizontal axis is obtained by linear discriminant analysis (LDA), while the vertical axis is a random projection.

image is watermarked. Similarly, if we can push a watermarked image away from this region, we will be able to falsely claim that it is not watermarked.

To show that such a region exists, we start by designing a simple experiment under idealized conditions, assuming only a single secret key. If there exists a latent region pertaining to a different set of watermarked images from each secret key, then we should be able to find latent directions that can lead randomly sampled latent vectors into these regions and thus be authenticated as being watermarked. To demonstrate this, we sampled 1,000 watermarked images from the Tree-Ring watermark using a specific secret key, along with the same number of non-watermarked images. We trained a linear support vector machine (SVM) model on this dataset to find such a direction in VAE's latent space (Colbois et al., 2021). We observed that simply traversing the latent space in the direction normal to the learned hyperplane could lead a non-watermarked sample towards being classified as watermarked, as shown in Figure 3. We can also remove the watermark from a watermarked image by traversing the latent space in the opposite direction. We formally define this region in Definition 1.

**Definition 1 (Watermark Region)** *For a watermarking method $\mathcal{W}$, we define a watermark region as a region in the clean latent space of the latent diffusion model, as,*

$$Z_0^{(w)}(\mathcal{W}, k) = \left\{ \mathbf{z}_0 \in Z_0 \mid \mathcal{M}_{\mathcal{W}}(\mathcal{I}^-(\mathbf{z}_0), k) < \tau \right\},$$

*where $Z_0$ represents the clean latent space at $t = 0$, $\mathcal{I}^-$ is the DDIM inversion process, $\mathcal{M}_{\mathcal{W}}$ is the matching function used to verify the presence of a particular key, $\tau$ is the operating threshold of the watermarking scheme, and $k$ is the secret key that was embedded.*

*This represents all the points in the clean latent space that lead to the key $k$ that was embedded in the initial noise latent vector when inverted.*

We also visualize high linear separability between watermarked images and non-watermarked ones in Figure 4. Yet, this edit in itself is not close to a real-world scenario because we used multiple watermarked samples from the same key and because we could not maintain the semantic content and quality of the corresponding image. This preliminary experiment indicates that such a watermarked region as defined in Definition 1 may exist, and it sets the stage for our novel attack strategy, which we describe in the next subsection. Our strategy aims to introduce imperceptible changes to the image while pushing and pulling latent representations into and out of watermarked regions for forgery and removal, respectively.

### 4.3 Imperceptible and Lightweight Attack Against Watermarking Methods

**Forgery Attack.** Based on the above intuition, to forge a watermark, the attacker needs to adversarially perturb a non-watermarked image so that it gets embedded in the watermarked region (Definition 1) of the clean latent space. However, finding and defining this region may require multiple generations from the same initial noise vector that has the secret key embedded in it. This is not feasible in scenarios where a model owner can regenerate the key every time (Arabi et al., 2024).

Instead, we propose a method that forges a watermark into the non-watermarked image by guiding it into the watermarked region using only a single watermarked image so that we minimize the distance between the latent representation of a non-watermarked image and that of a watermarked image while regularizing for content preservation. Our method utilizes the encoder of an off-the-shelf VAE $\mathcal{E}_{\phi}$, which was trained on a similar dataset (which can be different from the VAE of the diffusion model $\mathcal{E}_{\theta}$), to adversarially perturb a non-watermarked image.

The objective for finding the perturbation is defined as:

$$\min_{\boldsymbol{\delta}} \|\mathcal{E}_{\phi}(\mathbf{x}^{(c)} + \boldsymbol{\delta}) - \mathcal{E}_{\phi}(\mathbf{x}^{(w)})\|_2 + \lambda \|\boldsymbol{\delta}\|_2, \tag{3}$$

where $\boldsymbol{\delta}$ is the adversarial perturbation and $\lambda$ controls the trade-off between the strength of the perturbation and image content preservation. In Appendix A, we present an ablation study on the loss function design, comparing with other possible designs such as progressive gradient descent (PGD) (Kurakin et al., 2016; 2018).

**Removal Attack.** We adopt a similar approach to remove a watermark by adversarially perturbing a watermarked image so that we can ensure that it gets outside of the vulnerable watermarked region.

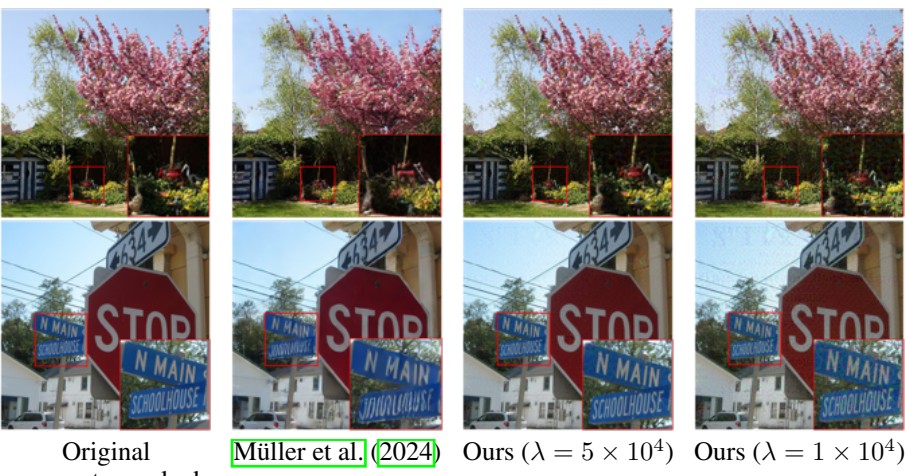

| Original non-watermarked | Müller et al. (2024) | Ours ($\lambda = 5 \times 10^4$) | Ours ($\lambda = 1 \times 10^4$) |

Figure 5: Visual comparison of our forgery attacks on the Tree-Ring watermarking method.

Table 1: Comparison with baselines on the attack success rate (ASR) and imperceptibility of the attack as judged using the $l_2$, $l_\infty$ distances and the LPIPS (Zhang et al., 2018), SSIM (Wang et al., 2004) and PSNR, when falsely watermarking non-watermarked images from the COCO2017 dataset assuming access to SDv1.4's VAE. The hyperparameter $\lambda$ in Equation 3 is set to $1 \times 10^4$.

| Method | Model | Method | ASR | $l_2$ | $l_\infty$ | LPIPS | SSIM | PSNR |
|--------|-------|--------|-----|-------|-----------|-------|------|------|
| Tree-Ring | SDv1.4 | Yang et al. (2024a) | 0.0 | 73.69 | **0.29** | **0.02** | **0.94** | 27.61 |
| | | Müller et al. (2024) | **100.0** | 115.22 | 1.51 | 0.13 | 0.70 | 23.03 |
| | | Ours | 91.06 | **63.22** | 1.10 | 0.33 | 0.76 | **28.87** |
| | SDv2.0 | Yang et al. (2024a) | 0.0 | 68.25 | **0.28** | **0.04** | **0.94** | 28.28 |
| | | Müller et al. (2024) | **100.0** | 114.42 | 1.50 | 0.13 | 0.70 | 23.08 |
| | | Ours | 93.81 | **63.78** | 1.08 | 0.34 | 0.76 | **28.78** |
| Gaussian Shading | SDv1.4 | Yang et al. (2024a) | 0.0 | 107.05 | **0.32** | **0.04** | **0.92** | 24.37 |
| | | Müller et al. (2024) | **100.0** | 116.38 | 1.50 | 0.13 | 0.70 | 22.96 |
| | | Ours | 96.85 | **37.27** | 0.70 | 0.19 | **0.87** | **33.48** |
| | SDv2.0 | Yang et al. (2024a) | 0.0 | 83.51 | **0.33** | **0.05** | **0.94** | 26.52 |
| | | Müller et al. (2024) | **100.0** | 116.47 | 1.49 | 0.13 | 0.70 | 22.95 |
| | | Ours | **100.0** | **36.78** | 0.66 | 0.19 | **0.87** | **33.60** |

Here, instead of using a real camera-captured non-watermarked image, we propose using a plain image whose pixel values are all set to the mean of the watermarked image $\mathbf{x}^{(w)}$. We do so because (i) the average image is naturally non-watermarked and does not rely on any specific external image for guidance and (ii) real images contain their own high-frequency information, which can lead to larger perturbations in the optimized adversarial image. We summarize the removal objective as:

$$\min_{\boldsymbol{\delta}} \|\mathcal{E}_\phi(\mathbf{x}^{(w)} + \boldsymbol{\delta}) - \mathcal{E}_\phi(\boldsymbol{\mu}_{\mathbf{x}^{(w)}})\|_2 + \lambda\|\boldsymbol{\delta}\|_2, \tag{4}$$

where $\boldsymbol{\mu}_{\mathbf{x}^{(w)}}$ is the plain image with all pixel values equal to the mean of the watermarked image $\mathbf{x}^{(w)}$. In Appendix C, we present an ablation study that justifies this design choice.

## 5 EXPERIMENTAL RESULTS

In this section, we showcase the effectiveness of our approach. We consider two attack scenarios, where the attacker has access to either (i) the VAE of the watermarked diffusion model or (ii) a proxy VAE that was trained on a similar dataset.

**Experimental Setup.** We consider two diffusion models, namely Stable Diffusion v1.4 (SDv1.4) and Stable Diffusion v2.0 (SDv2.0), to generate images of size $512 \times 512$. We generate watermarked images using the prompts available in the *Gustavosta/Stable-Diffusion-Prompts*. When generating reference watermarked images for forgery attacks, we use simpler prompts from the

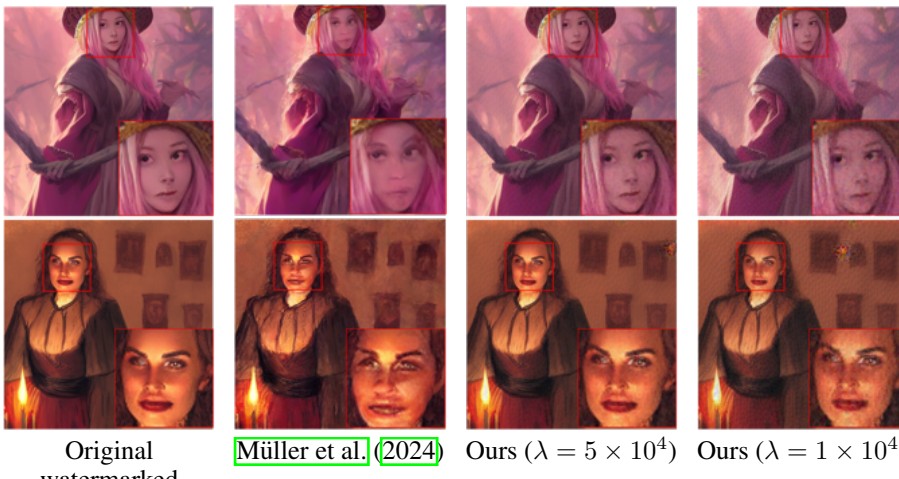

| Original | Müller et al. (2024) | Ours ($\lambda = 5 \times 10^4$) | Ours ($\lambda = 1 \times 10^4$) |
| watermarked | | | |

Figure 6: Qualitative comparison of our removal attack on the Tree-Ring watermarking method.

*runwayml-stable-diffusion-v1-5-eval-random-prompts* dataset, as the resultant images contain more visible watermark patterns/signal due to lower amounts of high-frequency information. We use the COCO2017 dataset (Lin et al., 2014) to obtain images without watermarks. We utilize the VAE from SDv1.4 to forge/remove a watermark generated from both SDv1.4 and SDv2.0, unless otherwise stated. When we use SDv2.0 to generate watermarked images, SDv1.4's VAE serves as a proxy VAE. We report results on the following publicly available watermarking systems that embed a key in the initial noise space, namely, Tree-Ring (Wen et al., 2023), RingID (Ci et al., 2024b), WIND (Arabi et al., 2024), and Gaussian Shading (Yang et al., 2024b). See Appendix H for further details.

**Evaluation Metrics.** We consider a forgery attack to be successful if the $p$-value statistical test comparing the extracted key and the secret key embedded in the reference watermarked image yields a $p$-value less than 0.05. For removal, we consider it a success if the $p$-value is greater than 0.05 with respect to the key in the original watermarked image. For Gaussian Shading, which uses a bit sequence, we compute the binary bit accuracy between the embedded and recovered keys. We report the average success rate across 200 examples while randomly drawing new non-watermarked and watermarked samples each time. Each watermarked sample is generated using a new random key.

Even though we test our attack in a black-box scenario, we do not make multiple attempts and consider it unsuccessful if the first attempt fails. We additionally report the $l_2$, $l_\infty$ distances and the Learned Perceptual Image Patch Similarity (LPIPS) (Zhang et al., 2018), Structural Similarity Index (SSIM) (Wang et al., 2004), and Peak Signal to Noise Ratio (PSNR) metrics between the original image and the adversarially perturbed image to assess the extent to which we alter the original image.

**Results - Forgery.** We summarize quantitative results in Table 1 for Tree-Ring and Gaussian Shading, comparing our attack with baselines. In Appendix Table 3, we present results on the trade-off between imperceptibility and attack success rate as well as results for RingID and WIND watermarks. Our method achieves performance comparable to that of Müller et al. (2024) at half the $l_2$ distance. As shown in Figure 5, our method better preserves the image content compared to Müller et al. (2024), since we perturb the denoised latent space with regularization. Yang et al. (2024a) achieves 0% ASR because it assumes that all images come from the same secret key, which is not the case in our experimental setting, where a model owner can regenerate the secret key with every generation. We showcase more qualitative examples in the Appendix I. We also show results on using the VAE from FLUX.1-dev in Appendix Table 8. Note that the more similar the encoder is to the one used by the model, the less noise is required.

**Results - Removal.** We report results on watermark removal in Table 2 for Tree-Ring and Gaussian Shading, comparing with baselines, and in Appendix Table 4 for RingID and WIND watermarks. Although our approach achieves success in removing the Tree-Ring watermark, it was harder to remove the watermark signal from RingID and WIND, as these methods embed a watermark into the entire initial latent noise space, which introduces a larger amount of watermark signal into images (see Section 6). We show qualitative results when removing the Tree-Ring watermark in Figure 6.

Table 2: Comparison with baselines on watermark removal on the attack success rate (ASR) and the imperceptibility of the changes using the $l_2$ distance and the LPIPS Zhang et al. (2018), SSIM Wang et al. (2004), and PSNR metrics. The hyperparameter $\lambda$ in Equation 4 is set to $1 \times 10^4$.

| Method | Model | Method | ASR | $l_2$ | LPIPS | SSIM | PSNR |
|---|---|---|---|---|---|---|---|
| Tree-Ring | SDv1.4 | Yang et al. (2024a) | 1.15 | 115.06 | **0.09** | **0.90** | 19.97 |
| | | Zhao et al. (2025) | 5.05 | 148.71 | 0.18 | 0.71 | 17.73 |
| | | Müller et al. (2024) | **100.0** | 391.50 | 0.64 | 0.46 | 12.92 |
| | | Kassis and Hengartner (2024) | 39.5 | 423.12 | 0.47 | 0.39 | 11.57 |
| | | An et al. (2024) | 5.5 | 244.52 | 0.43 | 0.63 | 13.58 |
| | | Ours | 98.84 | **62.87** | 0.30 | 0.78 | **28.91** |
| | SDv2.0 | Yang et al. (2024a) | 2.60 | 118.28 | **0.12** | **0.88** | 19.42 |
| | | Zhao et al. (2025) | 6.0 | 142.63 | 0.17 | 0.69 | 18.80 |
| | | Müller et al. (2024) | **100.0** | 418.21 | 0.68 | 0.39 | 12.39 |
| | | Kassis and Hengartner (2024) | 39.0 | 439.45 | 0.48 | 0.36 | 11.54 |
| | | An et al. (2024) | 5.5 | 242.70 | 0.43 | 0.64 | 13.88 |
| | | Ours | 98.36 | **67.62** | 0.30 | 0.77 | **28.02** |
| Gaussian Shading | SDv1.4 | Yang et al. (2024a) | 4.0 | 131.43 | **0.07** | **0.89** | 19.78 |
| | | Zhao et al. (2025) | 3.0 | 114.86 | 0.13 | 0.74 | 20.77 |
| | | Müller et al. (2024) | **100.0** | 499.68 | 0.74 | 0.31 | 10.87 |
| | | Kassis and Hengartner (2024) | **100.0** | 376.52 | 0.47 | 0.39 | 12.96 |
| | | An et al. (2024) | 4.0 | 200.43 | 0.38 | 0.74 | 15.58 |
| | | Ours | 70.10 | **74.10** | 0.29 | 0.77 | **24.12** |
| | SDv2.0 | Yang et al. (2024a) | 0.0 | 84.78 | **0.04** | **0.95** | 26.33 |
| | | Zhao et al. (2025) | 0.0 | 113.73 | 0.11 | 0.70 | 23.34 |
| | | Müller et al. (2024) | **100.0** | 524.10 | 0.77 | 0.28 | 10.44 |
| | | Kassis and Hengartner (2024) | **100.0** | 392.90 | 0.49 | 0.38 | 12.60 |
| | | An et al. (2024) | 7.0 | 208.10 | 0.38 | 0.72 | 15.33 |
| | | Ours | 59.13 | **68.73** | 0.27 | 0.77 | **28.13** |

**Results - Computational time.** Our attack takes 7.82 minutes per image while Müller et al. (2024) - the only other successful forgery attack - takes 12.29 minutes on one A100 GPU on average across 100 attacks.

**Results - Ablation Studies on Hyperparameter Values.** We sweep the hyperparameter $\lambda$ to see the trade-off between the attack success rate (ASR) and imperceptibility of watermarking attacks. We report the experimental results in Tables 3 and 4, which show that a larger $\lambda$ value (a stronger regularization) leads to a better preservation of image content at the cost of a lower ASR.

**Results - Ablation Studies on the design of the loss function.** We show experimental results on alternative loss function design. We specifically test our proposed loss function with using a progressive gradient descent and version of it in Appendix A.

## 6  DISCUSSION AND LIMITATION

**Multi-Pattern vs. Single-Pattern Watermarks.** We found that forging was generally easier for our method than watermark removal across all approaches, particularly for RingID, and WIND watermarks, where our attack was unsuccessful in removing the watermark signal. We suggest that this is because these methods do not merely encode a single pattern, like Tree-Ring, but also encode additional information, such as the model owner's identity or other metadata. As the number of possible embedded patterns increases, more encoded information is required to correctly identify the pattern. This, in turn, necessitates a stronger signal-to-noise ratio (Shannon, 1949), where noise refers to patterns in the initial noise that are unrelated to our watermark. When more signal is associated with the embedded watermark, forging at least part of it becomes easier, while completely removing it becomes more difficult. In contrast, Tree-Ring embeds a simpler pattern that affects only a portion of the initial noise, making it harder to forge within a fixed budget. Using higher detection thresholds results in the opposite: forgery becomes harder and removal easier.

## 7  CONCLUSION

In this paper, we expose a vulnerability of initial noise-based watermarking schemes for diffusion models. We show that when a watermark key is embedded in the initial noise, a latent watermarked region may form in the denoised latent space. This makes it easier for an attacker to forge the

Table 3: Forgery attack performance trade-off between attack success rate (ASR) and imperceptibility of the attack as judged using the $l_2$, $l_\infty$ distances and the Learned Perceptual Image Patch Similarity (LPIPS) (Zhang et al., 2018), Structural Similarity Index (SSIM) (Wang et al., 2004), and Peak Signal to Noise Ratio (PSNR) metrics. We use non-watermarked images from the COCO2017 dataset and employ the VAE from SDv1.4 for optimization. The hyperparameter $\lambda$ in Equation 3 controls the trade-off between ASR and the amount of perturbation we introduce.

| Method | Model | $\lambda$ | ASR | $l_2$ | $l_\infty$ | LPIPS | SSIM | PSNR |
|---|---|---|---|---|---|---|---|---|
| Tree-Ring (Wen et al., 2023) | SDv1.4 | $5 \times 10^4$ | 78.65 | 33.90 | 0.69 | 0.17 | 0.89 | 34.32 |
| | | $2 \times 10^4$ | 86.93 | 48.42 | 0.89 | 0.26 | 0.82 | 31.20 |
| | | $1 \times 10^4$ | 91.06 | 63.22 | 1.10 | 0.33 | 0.76 | 28.87 |
| | SDv2.0 | $5 \times 10^4$ | 79.89 | 34.09 | 0.69 | 0.17 | 0.88 | 34.26 |
| | | $2 \times 10^4$ | 90.72 | 48.83 | 0.91 | 0.26 | 0.82 | 31.11 |
| | | $1 \times 10^4$ | 93.81 | 63.78 | 1.08 | 0.34 | 0.76 | 28.78 |
| | FLUX.1-dev | $5 \times 10^4$ | 45.95 | 92.82 | 0.74 | 0.27 | 0.81 | 18.71 |
| | | $2 \times 10^4$ | 57.44 | 109.36 | 0.89 | 0.36 | 0.74 | 18.35 |
| | | $1 \times 10^4$ | 64.64 | 118.91 | 1.05 | 0.44 | 0.67 | 18.92 |
| RingID (Ci et al., 2024b) | SDv1.4 | $5 \times 10^4$ | 100.0 | 38.45 | 0.68 | 0.20 | 0.87 | 33.21 |
| | | $2 \times 10^4$ | 100.0 | 55.20 | 0.86 | 0.30 | 0.80 | 30.06 |
| | | $1 \times 10^4$ | 100.0 | 73.08 | 1.03 | 0.38 | 0.73 | 27.63 |
| | SDv2.0 | $5 \times 10^4$ | 100.0 | 37.31 | 0.66 | 0.19 | 0.87 | 33.48 |
| | | $2 \times 10^4$ | 100.0 | 53.94 | 0.84 | 0.29 | 0.80 | 30.27 |
| | | $1 \times 10^4$ | 100.0 | 71.53 | 1.00 | 0.37 | 0.73 | 27.82 |
| WIND (Arabi et al., 2024) | SDv1.4 | $5 \times 10^4$ | 97.56 | 38.82 | 0.70 | 0.20 | 0.87 | 33.11 |
| | | $2 \times 10^4$ | 97.56 | 56.13 | 0.89 | 0.29 | 0.80 | 29.88 |
| | | $1 \times 10^4$ | 97.56 | 74.66 | 1.06 | 0.38 | 0.73 | 27.38 |
| | SDv2.0 | $5 \times 10^4$ | 100.0 | 37.47 | 0.67 | 0.19 | 0.87 | 33.45 |
| | | $2 \times 10^4$ | 100.0 | 54.18 | 0.84 | 0.28 | 0.80 | 30.23 |
| | | $1 \times 10^4$ | 100.0 | 71.86 | 0.99 | 0.37 | 0.74 | 27.78 |
| Gaussian Shading (Yang et al., 2024b) | SDv1.4 | $5 \times 10^4$ | 96.85 | 37.27 | 0.70 | 0.19 | 0.87 | 33.48 |
| | | $2 \times 10^4$ | 96.96 | 54.00 | 0.88 | 0.29 | 0.80 | 30.21 |
| | | $1 \times 10^4$ | 96.96 | 71.97 | 1.05 | 0.37 | 0.73 | 27.64 |
| | SDv2.0 | $5 \times 10^4$ | 100.0 | 36.78 | 0.66 | 0.19 | 0.87 | 33.60 |
| | | $2 \times 10^4$ | 100.0 | 52.99 | 0.85 | 0.29 | 0.80 | 30.42 |
| | | $1 \times 10^4$ | 100.0 | 69.83 | 1.02 | 0.37 | 0.74 | 28.02 |

Table 4: Watermark removal attack performance trade-off between attack success rate (ASR) and imperceptibility of the attack as judged using the $l_2$, $l_\infty$ distances and the Learned Perceptual Image Patch Similarity (LPIPS) (Zhang et al., 2018), Structural Similarity Index (SSIM) (Wang et al., 2004), and Peak Signal to Noise Ratio (PSNR) metrics. We use the VAE from SDv1.4 for optimization. The hyperparameter $\lambda$ in Equation 4 controls the trade-off between ASR and the amount of perturbation we introduce.

| Method | Model | $\lambda$ | ASR | $l_2$ | $l_\infty$ | LPIPS | SSIM | PSNR |
|---|---|---|---|---|---|---|---|---|
| Tree-Ring (Wen et al., 2023) | SDv1.4 | $5 \times 10^4$ | 94.21 | 34.13 | 0.87 | 0.15 | 0.89 | 34.23 |
| | | $2 \times 10^4$ | 97.68 | 47.93 | 1.09 | 0.23 | 0.83 | 31.27 |
| | | $1 \times 10^4$ | 98.84 | 62.87 | 1.31 | 0.30 | 0.78 | 28.91 |
| | SDv2.0 | $5 \times 10^4$ | 95.08 | 40.88 | 0.98 | 0.15 | 0.88 | 31.71 |
| | | $2 \times 10^4$ | 97.80 | 53.56 | 1.15 | 0.23 | 0.82 | 29.82 |
| | | $1 \times 10^4$ | 98.36 | 67.62 | 1.37 | 0.30 | 0.77 | 28.02 |
| | FLUX.1-dev | $5 \times 10^4$ | 0.5 | 30.27 | 0.87 | 0.025 | 0.98 | 35.06 |
| | | $2 \times 10^4$ | 0.5 | 63.24 | 1.49 | 0.069 | 0.96 | 28.89 |
| | | $1 \times 10^4$ | 0.5 | 97.16 | 1.88 | 0.13 | 0.93 | 25.19 |
| RingID (Ci et al., 2024b) | SDv1.4 | $5 \times 10^4$ | 0.0 | 35.62 | 0.90 | 0.14 | 0.88 | 33.87 |
| | | $2 \times 10^4$ | 0.0 | 50.07 | 1.13 | 0.22 | 0.83 | 30.90 |
| | | $1 \times 10^4$ | 0.0 | 65.80 | 1.33 | 0.29 | 0.77 | 28.53 |
| | | Muller (Steps=50) | 29.0 | 118.41 | 1.52 | 0.15 | 0.66 | 22.98 |
| | | Muller (Steps=10) | 0.0 | 55.84 | 1.11 | 0.04 | 0.87 | 29.44 |
| | SDv2.0 | $5 \times 10^4$ | 0.0 | 36.46 | 0.92 | 0.14 | 0.88 | 33.67 |
| | | $2 \times 10^4$ | 0.0 | 51.69 | 1.15 | 0.21 | 0.82 | 30.63 |
| | | $1 \times 10^4$ | 0.0 | 68.03 | 1.37 | 0.28 | 0.76 | 28.24 |
| WIND (Arabi et al., 2024) | SDv1.4 | $5 \times 10^4$ | 0.0 | 48.11 | 1.06 | 0.15 | 0.87 | 29.59 |
| | | $2 \times 10^4$ | 0.0 | 60.46 | 1.21 | 0.23 | 0.81 | 28.30 |
| | | $1 \times 10^4$ | 0.0 | 74.47 | 1.37 | 0.29 | 0.75 | 26.90 |
| | SDv2.0 | $5 \times 10^4$ | 0.0 | 35.99 | 0.93 | 0.14 | 0.89 | 33.79 |
| | | $2 \times 10^4$ | 0.0 | 50.79 | 1.16 | 0.21 | 0.83 | 30.79 |
| | | $1 \times 10^4$ | 0.0 | 66.86 | 1.37 | 0.28 | 0.77 | 28.40 |
| Gaussian Shading (Yang et al., 2024b) | SDv1.4 | $5 \times 10^4$ | 11.41 | 34.68 | 0.86 | 0.13 | 0.89 | 33.95 |
| | | $2 \times 10^4$ | 39.79 | 63.76 | 1.15 | 0.23 | 0.81 | 23.99 |
| | | $1 \times 10^4$ | 70.10 | 74.10 | 1.31 | 0.29 | 0.77 | 24.12 |
| | SDv2.0 | $5 \times 10^4$ | 12.23 | 38.21 | 0.98 | 0.13 | 0.88 | 33.17 |
| | | $2 \times 10^4$ | 34.73 | 53.55 | 1.19 | 0.21 | 0.83 | 30.26 |
| | | $1 \times 10^4$ | 59.13 | 68.73 | 1.39 | 0.27 | 0.77 | 28.13 |

watermark by perturbing a sample so that it lies within this region. We show that a similar approach can also be used for removal by pushing a watermarked latent away from this region. We hope this work motivates future research on improving watermarking systems in the face of adversaries.

ETHICS STATEMENT

Watermarking methods are widely used for ensuring ethical use of media content, as they allow easy authentication of (a) who created the content and (b) whether or not the content was tampered with. The same is applied to generative models to verify whether or not an image was generated and, if so, by which model. This makes it vital to ensure that these systems are robust to adversaries. This paper exposes a vulnerability in such watermarking methods for diffusion models, with the hope that it will allow researchers to understand the current limitations of watermarking approaches and aid them in developing more robust ones.

REPRODUCIBILITY STATEMENT

To facilitate reproducing the results from the paper, we have included the codebase of the paper in the supplementary materials. Further, we have provided all necessary implementation details, including hyperparameters, in the main paper and the Appendix.

THE USE OF LARGE LANGUAGE MODELS (LLMS)

We have NOT utilized Large Language Models to (i) help with coming up with ideas for the paper, (ii) help with writing the code, or (iii) help with paper writing or polishing.

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

## A    EXPERIMENTAL RESULTS USING ALTERNATIVE LOSS FORMULATIONS

In this section, we discuss an alternative design of the adversarial loss function to forge a watermark signal.

### A.1    ALTERNATIVE LOSS FORMULATIONS

**Using Progressive Gradient Descent**    In our main experiments, we control the amount of perturbations by adding a term to our loss function to minimize the perturbations while attacking the watermarking method (Equation 3). In this section, we discuss an alternative optimization objective by utilizing progressive gradient descent (Kurakin et al., 2016; 2018), through which we can set an explicit perturbation budget to control the amount of perturbations we introduce.

The objective for the forgery attack in this case becomes,

$$\min_{\boldsymbol{\delta}} \|\mathcal{E}_\phi(\mathbf{x}^{(c)} + \boldsymbol{\delta}) - \mathcal{E}_\phi(\mathbf{x}^{(w)})\|_2 \quad \text{s.t.} \quad \|\boldsymbol{\delta}\|_\infty \leq \epsilon, \tag{5}$$

where $\epsilon$ is the perturbation budget.

**Perturbations in the Frequency Domain**    Furthermore, we can reduce the perceptible impact by, instead of directly perturbing in the RGB space, converting the image to its frequency domain using the discrete cosine transform (DCT) and perturbing only the high-frequency regions in the frequency domain (Jia et al., 2022; Luo et al., 2022). To control the frequency regions we want to alter, we introduce a binary mask $\mathbf{m} \in \{0,1\}^{N \times N}$, which consists of zeros in the upper-left triangle that corresponds to the low-frequency region ending at indices $\lfloor (1-\alpha) \times N \rfloor$ and ones everywhere else. Here, $\alpha \in [0,1]$ controls the frequency regions we want to perturb (see Figure 7 for reference on how the mask looks).

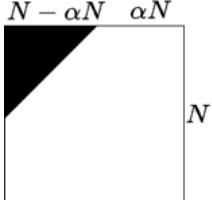

Figure 7: Visualization of the mask for controlling the adversarial perturbations. Noise is added only in the higher frequency region of an image.

We can write this objective as,

$$\min_{\boldsymbol{\delta}} \|\mathcal{E}_\phi(\text{IDCT}(\text{DCT}(\mathbf{x}^{(c)}) + \mathbf{m} \odot \boldsymbol{\delta})) - \mathcal{E}_\phi(\mathbf{x}^{(w)})\|_2 \quad \text{s.t.} \quad \|\boldsymbol{\delta}\|_\infty \leq \epsilon. \tag{6}$$

### A.2    EXPERIMENTAL RESULTS

We summarize the results in Table 5, where we show that using the proposed loss formulation in Equation 3 achieves better imperceptibility results with a similar attack success rate. We also show visual examples in Figure 8. Although the optimization using Equation 6 yields lower imperceptibility scores, it can better preserve lower-frequency regions, resulting in improved visual results.

Table 5: Comparing trade-off between attack success rate (ASR) and imperceptibility of the attack for different optimization objectives when they achieve similar ASR. We have set the hyperparameters $\epsilon$ in Eq. 5 to 0.1, $(\epsilon, \alpha N)$ in Eq. 6 to (3, 300) and $\lambda$ in Eq. 3 to $2.5 \times 10^4$.

| Method | ASR | $l_2$ | $l_\infty$ | LPIPS | SSIM | PSNR |
|--------|-----|-------|------------|-------|------|------|
| Eq. 5 | 83.24 | 68.04 | 0.10 | 0.35 | 0.66 | 28.31 |
| Eq. 6 | 83.33 | 67.77 | 0.93 | 0.32 | 0.66 | 28.35 |
| Eq. 3 | 84.91 | 44.34 | 0.85 | 0.23 | 0.84 | 31.97 |

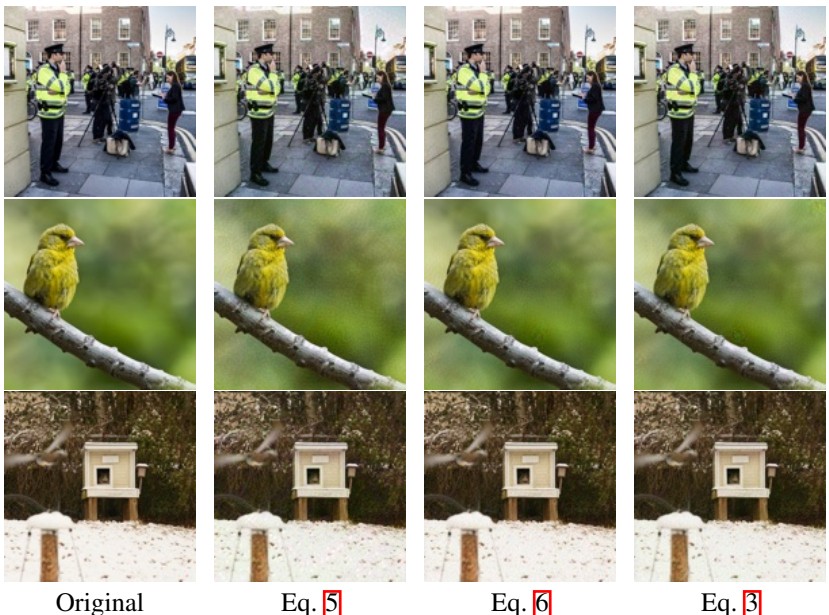

| Original | Eq. 5 | Eq. 6 | Eq. 3 |

Figure 8: Comparing trade-off between attack success rate (ASR) and imperceptibility of the attack for different optimization objectives when they achieve similar ASR.

## B   WATERMARK REMOVAL USING IMAGES WITH FIXED PIXEL VALUES

In Section 4.3 for watermark removal, we use images with all pixel values equal to the mean value of the given watermarked image for guidance. In this section, we demonstrate that this approach works better in practice compared to using a fixed pixel value of 127.5. We report results in Table 6. As shown, using the mean value performs better using a fixed pixel value of 127.5.

Table 6: Comparing trade-off between attack success rate (ASR) and imperceptibility of the watermark removal attack when using either images with all pixel values equal to 127.5 or the mean of the watermarked image for guidance. These experiments are run on SDv1.4 for the Gaussian Shading watermarking scheme. The hyperparameter $\lambda$ in Equation 4 is set to $1 \times 10^4$.

| Method | ASR | $l_2$ | $l_\infty$ | LPIPS | SSIM | PSNR |
|--------|-----|-------|-----------|-------|------|------|
| Mean | 70.10 | 74.10 | 1.31 | 0.29 | 0.77 | 24.12 |
| 127.5 | 64.58 | 78.44 | 1.36 | 0.31 | 0.74 | 23.18 |

## C   WATERMARK REMOVAL USING REAL IMAGES

Another option is to directly use a camera-captured non-watermarked image for guidance. In this section, we show that our approach works better in practice without requiring a non-watermarked image. For the study, we use non-watermarked images from the COCO (Lin et al., 2014) dataset for guidance, where we minimize the distance between their respective representations while perturbing the watermarked image. The optimization objective remains the same as Equation 4, i.e.,

$$\min_{\boldsymbol{\delta}} \|\mathcal{E}_\phi(\mathbf{x}^{(w)} + \boldsymbol{\delta}) - \mathcal{E}_\phi(\mathbf{x}^{(c)})\|_2 + \lambda\|\boldsymbol{\delta}\|_2, \tag{7}$$

where $\mathbf{x}^{(c)}$ is a randomly selected real image. We show results when using a real image in Table 7. We also show qualitative comparisons in Figure 9 and Figure 10.

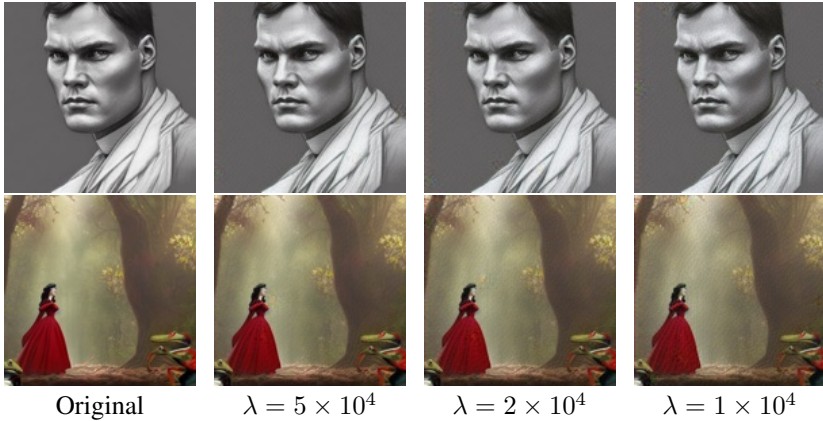

Original  $\lambda = 5 \times 10^4$  $\lambda = 2 \times 10^4$  $\lambda = 1 \times 10^4$

Figure 9: Examples showing successful watermark removal attacks on the Tree-Ring watermarking method with different hyperparameter $\lambda$ values when using *an image with all pixels equal to the mean of the watermarked image for guidance*.

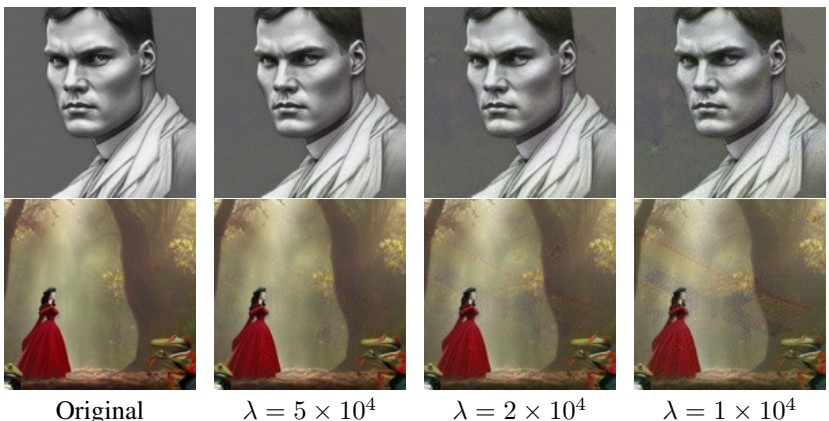

Original  $\lambda = 5 \times 10^4$  $\lambda = 2 \times 10^4$  $\lambda = 1 \times 10^4$

Figure 10: Examples showing successful watermark removal attacks on the Tree-Ring watermarking method with different hyperparameter $\lambda$ values when using *real images for guidance*.

## D  EXPERIMENTAL RESULTS ON USING A SIGNIFICANTLY DIFFERENT VAE

We conducted a study to assess whether our forgery attack is still successful if the VAE learns a different representation. For this experiment, we have chosen the VAE from FLUX.1-dev (Black Forest Labs, 2024), which uses a different compression ratio as compared to SDv1.4. The FLUX.1-dev VAE compresses the latent representation to 16 channels as compared to 4 by SDv1.4. This implies that there are differences in the latent representations learned by these models. We present results in Table 8, which shows that for RingID and WIND, we can preserve image quality (PSNR≈35) while achieving a high attack success rate. Tree-Ring, which embeds a smaller signal into the initial latent noise space, was harder to forge using a FLUX.1-dev VAE. This observation can be interesting to inform future research on developing robust watermarking methods. A potential defense based on this observation could be for a model owner to train a dissimilar VAE from publicly available ones so as not to allow such attacks to be successful.

Table 7: Results on watermark removal using a real image for guidance (Equation 7).

| Method | Model | $\lambda$ | ASR | $l_2$ | $l_\infty$ | LPIPS | SSIM | PSNR |
|---|---|---|---|---|---|---|---|---|
| Tree-Ring (Wen et al., 2023) | SDv1.4 | $5 \times 10^4$ | 68.23 | 41.19 | 0.79 | 0.21 | 0.86 | 31.95 |
| | | $2 \times 10^4$ | 78.94 | 61.04 | 0.97 | 0.31 | 0.78 | 28.86 |
| | | $1 \times 10^4$ | 83.62 | 80.81 | 1.09 | 0.41 | 0.70 | 26.73 |
| | SDv2.0 | $5 \times 10^4$ | 61.62 | 44.40 | 0.88 | 0.21 | 0.86 | 31.60 |
| | | $2 \times 10^4$ | 70.81 | 63.13 | 1.04 | 0.32 | 0.77 | 28.68 |
| | | $1 \times 10^4$ | 74.59 | 84.01 | 1.15 | 0.42 | 0.69 | 26.35 |
| RingID (Ci et al., 2024b) | SDv1.4 | $5 \times 10^4$ | 0.0 | 38.25 | 0.74 | 0.18 | 0.87 | 33.27 |
| | | $2 \times 10^4$ | 1.10 | 56.98 | 0.95 | 0.28 | 0.79 | 29.79 |
| | | $1 \times 10^4$ | 1.65 | 78.01 | 1.13 | 0.37 | 0.72 | 27.05 |
| | SDv2.0 | $5 \times 10^4$ | 0.0 | 38.90 | 0.73 | 0.18 | 0.87 | 33.13 |
| | | $2 \times 10^4$ | 0.54 | 58.26 | 0.97 | 0.28 | 0.79 | 29.60 |
| | | $1 \times 10^4$ | 0.54 | 79.85 | 1.15 | 0.36 | 0.72 | 26.85 |
| WIND (Arabi et al., 2024) | SDv1.4 | $5 \times 10^4$ | 0.0 | 50.64 | 0.98 | 0.19 | 0.85 | 29.33 |
| | | $2 \times 10^4$ | 1.09 | 66.79 | 1.07 | 0.29 | 0.78 | 27.66 |
| | | $1 \times 10^4$ | 1.09 | 85.81 | 1.23 | 0.38 | 0.70 | 25.85 |
| | SDv2.0 | $5 \times 10^4$ | 0.0 | 38.44 | 0.74 | 0.18 | 0.87 | 33.23 |
| | | $2 \times 10^4$ | 0.54 | 57.55 | 0.97 | 0.28 | 0.80 | 29.71 |
| | | $1 \times 10^4$ | 0.54 | 78.91 | 1.13 | 0.36 | 0.72 | 26.96 |
| Gaussian Shading (Yang et al., 2024b) | SDv1.4 | $5 \times 10^4$ | 28.0 | 46.65 | 0.84 | 0.21 | 0.84 | 28.36 |
| | | $2 \times 10^4$ | 65.0 | 73.49 | 1.08 | 0.32 | 0.75 | 23.87 |
| | | $1 \times 10^4$ | 81.0 | 92.20 | 1.21 | 0.42 | 0.67 | 23.70 |
| | SDv2.0 | $5 \times 10^4$ | 25.0 | 44.31 | 0.84 | 0.18 | 0.86 | 31.58 |
| | | $2 \times 10^4$ | 49.0 | 65.25 | 1.06 | 0.29 | 0.78 | 28.44 |
| | | $1 \times 10^4$ | 74.0 | 87.61 | 1.20 | 0.39 | 0.70 | 25.96 |

Table 8: Comparing forgery performance when using a significantly different VAE. We use the VAEs from SDv1.4 and FLUX.1-dev to attack watermarks in SDv1.4.

| Method | VAE | $\lambda$ | ASR | $l_2$ | $l_\infty$ | LPIPS | SSIM | PSNR |
|---|---|---|---|---|---|---|---|---|
| Tree-Ring | FLUX.1-dev | $1 \times 10^3$ | 2.63 | 46.51 | 1.42 | 0.26 | 0.84 | 31.53 |
| | | $5 \times 10^2$ | 2.94 | 52.46 | 1.51 | 0.29 | 0.82 | 30.50 |
| | SDv1.4 | $5 \times 10^4$ | 78.65 | 33.90 | 0.69 | 0.17 | 0.89 | 34.32 |
| | | $1 \times 10^4$ | 91.06 | 63.22 | 1.10 | 0.33 | 0.76 | 28.87 |
| RingID | FLUX.1-dev | $1 \times 10^4$ | 61.20 | 22.70 | 1.08 | 0.12 | 0.93 | 37.69 |
| | | $5 \times 10^3$ | 82.51 | 29.22 | 1.08 | 0.17 | 0.91 | 35.58 |
| | SDv1.4 | $5 \times 10^4$ | 100.0 | 38.45 | 0.68 | 0.20 | 0.87 | 33.21 |
| | | $1 \times 10^4$ | 100.0 | 73.08 | 1.03 | 0.38 | 0.73 | 27.63 |
| WIND | FLUX.1-dev | $1 \times 10^4$ | 55.72 | 22.83 | 1.04 | 0.12 | 0.94 | 37.51 |
| | | $5 \times 10^3$ | 84.44 | 30.83 | 1.14 | 0.17 | 0.91 | 34.76 |
| | SDv1.4 | $5 \times 10^4$ | 97.56 | 38.82 | 0.70 | 0.20 | 0.87 | 33.11 |
| | | $1 \times 10^4$ | 97.56 | 74.66 | 1.06 | 0.38 | 0.73 | 27.38 |
| Gaussian Shading | FLUX.1-dev | $1 \times 10^3$ | 19.69 | 47.43 | 1.37 | 0.27 | 0.83 | 31.27 |
| | | $5 \times 10^2$ | 20.20 | 54.14 | 1.50 | 0.31 | 0.80 | 30.13 |
| | SDv1.4 | $5 \times 10^4$ | 96.85 | 37.27 | 0.70 | 0.19 | 0.87 | 33.48 |
| | | $1 \times 10^4$ | 96.96 | 71.97 | 1.05 | 0.37 | 0.73 | 27.64 |

## E  DISCUSSION: BASELINE DISTORTION IN RINGID-TYPE METHODS

While evaluating image quality, we found that even unper-turbed images generated by the RingID and WIND methods exhibit some distortion, as shown in Figure 11. This issue was also discussed in Appendix C of the RingID manuscript (Ci et al., 2024b). The RingID method (and WIND, by incorporating RingID in one of its variants) mitigated these artifacts by embedding a watermark only in a single channel. However, in our evaluation, the dis-tortion is often still noticeable and is more pronounced in the case of simpler prompts. This further highlights the

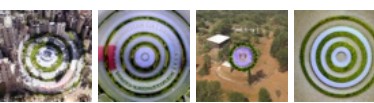

Figure 11: Examples of images gen-erated using RingID, where the water-mark signal is visible even in the final generated image.

strength of the watermark signal embedded by these techniques, making it harder to remove.

## F  DISCUSSION: ALTERNATIVE WATERMARKING APPROACHES

Other watermarking approaches that add a pattern during the latent decoding phase are less suscep-tible to attacks such as ours (Ci et al., 2024a; Fernandez et al., 2023). As these methods fine-tune the decoder, the adversary would require access to a similar decoder to attack the system. There is an inherent trade-off here, i.e., these methods are less resistant to image transformations, which was a major advantage of initial noise-based watermarking schemes.

Alternative approaches to building a more secure watermarking scheme could include embedding a secret message that encodes information pertaining to the contents of the watermarked image. This would make it harder for an attacker to forge a watermark, as they would need to embed a new message, which can only be done if they have access to both the entire diffusion model and the secret message generation method. It would allow a model owner to quickly verify that the content of the image and the recovered secret message match.

## G  DISCUSSION: TRADE-OFF BETWEEN LPIPS AND PSNR

In this section, we present examples to showcase that, for forgery and removal attacks such as those presented in this paper, $L_2$ distance and PSNR value are more important to optimize than LPIPS distance. Forgery and removal attacks are carried out on specific images, in which the attacker does not wish to change the content but rather only wishes to introduce or remove the watermark signal *with minimal changes to the semantic content*. As we show in the examples below, LPIPS is prone to showing low distance values even when the semantic content changes, as demonstrated by an LPIPS distance of 0.023 between Figures 12(a) and 12(b). In Figure 12(c), we have only added salt-and-pepper noise with probability 0.005, and it gives an LPIPS distance of 0.118 wrt to Figure 12(a), making it 5 times worse than Figure 12(b) in terms of LPIPS distance.

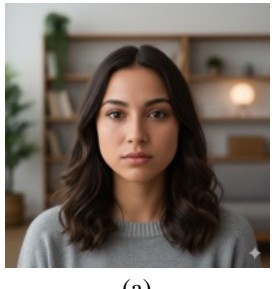 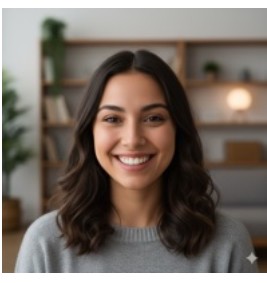 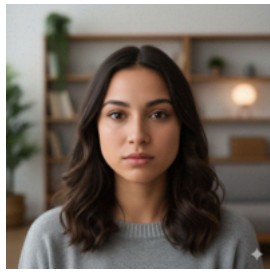

(a)      (b) LPIPS(a,b) = 0.023      (c) LPIPS(a,c) = 0.118

Figure 12: We showcase an example where LPIPS is extremely low even when the semantic content between the images changes, as seen in (b), but can be high even with very low salt-and-pepper noise (p=0.005), as seen in (c).

# H DETAILED EXPERIMENTAL SETUP

## H.1 PROMPT DATASETS

We utilize two datasets namely, the *Gustavosta/Stable-Diffusion-Prompts*[1] dataset and the *runwayml-stable-diffusion-v1-5-eval-random-prompts*[2] dataset. The former contains around 80,000 prompts extracted from the image finder for Stable Diffusion: "Lexica.art". These are generally longer and more detailed prompts. The latter, on the other hand, contains 200 short and simple prompts that contain less information/ details.

## H.2 WATERMARKING METHODS

We consider four latent-noise based watermarking schemes - Tree-Rings (Wen et al., 2023), RingID (Ci et al., 2024b), Gaussian Shading (Yang et al., 2024b) and WIND (Arabi et al., 2024). We utilized their publicly available implementations from the following GitHub repositories,

- Tree-Ring: https://github.com/YuxinWenRick/tree-ring-watermark.
- RingID: https://github.com/showlab/RingID.
- Gaussian Shading: https://github.com/bsmhmmlf/Gaussian-Shading/.
- WIND: https://github.com/anonymousiclr2025submission/Hidden-in-the-Noise.

## H.3 HYPERPARAMETERS

We adopt the following hyperparameter values in our optimization:

- Number of iterations: 15,000
- Learning rate $\alpha$: 0.020
- Image size: $512 \times 512$
- Stable Diffusion versions: CompVis/stable-diffusion-v1-4 and stabilityai/stable-diffusion-2

---

[1] https://huggingface.co/datasets/Gustavosta/Stable-Diffusion-Prompts
[2] https://huggingface.co/datasets/yuvalkirstain/runwayml-stable-diffusion-v1-5-eval-random-prompts

# I   MORE VISUAL EXAMPLES

## I.1   EXISTENCE OF LATENT DIRECTIONS

We present additional visual examples showcasing the effectiveness of latent directions in forging and removing watermarks in Figure 13.

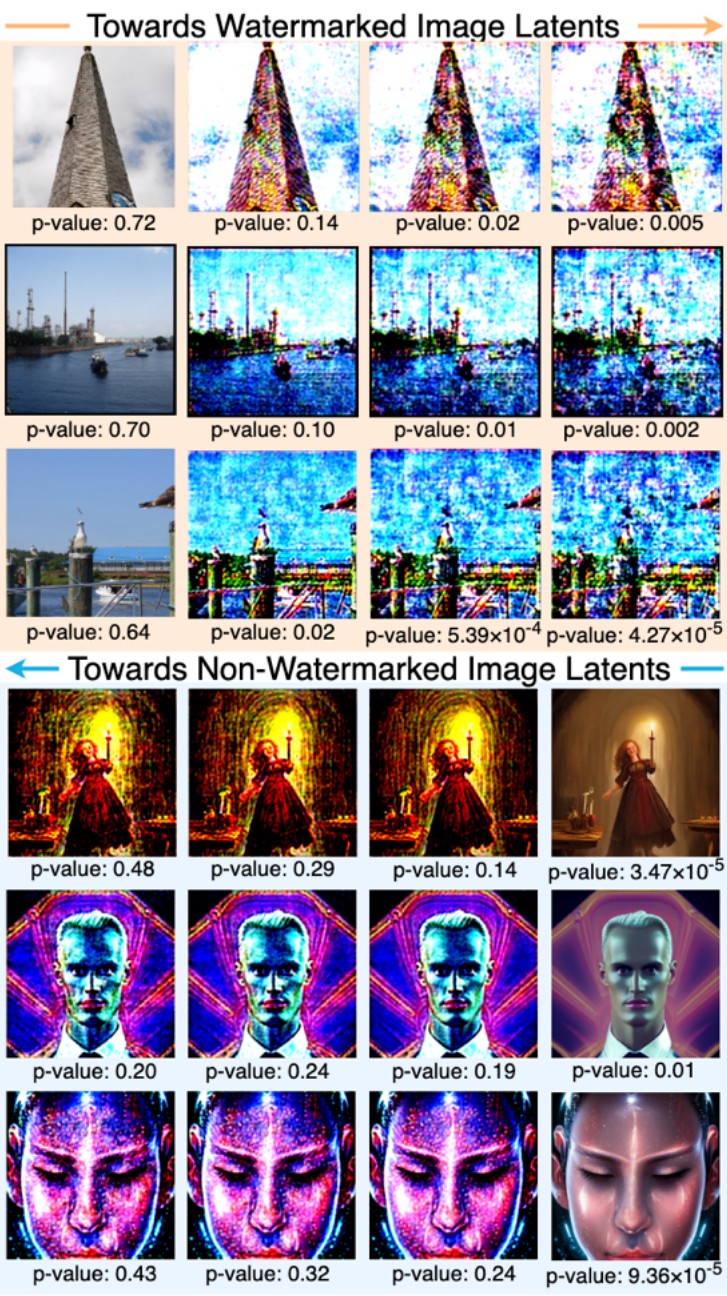

Figure 13: Examples showcasing the idea that there exist latent directions which pertain to watermarking and removal. We learn these directions using a linear SVM, and they are the normal to the learned hyperplane. Traversing further along them increases the strength of the attack.

## I.2 QUALITATIVE RESULTS ON WATERMARK FORGERY AND REMOVAL

We provide additional visual examples to show that successful watermark forgery and removal attacks do not harm the semantics or quality of the resultant image. We show examples of forgery attacks on Tree-Ring in Figure 15, RingID in Figure 16, WIND in Figure 17, and Gaussian Shading in Figure 18. We show examples of the removal attack in Figure 9 and Figure 14 for the Tree-Ring system and in Figure 19 for the Gaussian Shading system.

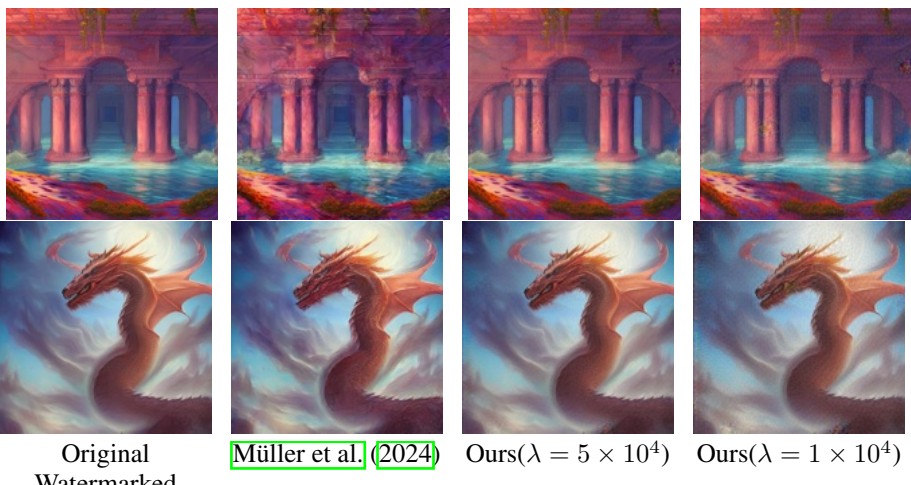

| Original Watermarked | Müller et al. (2024) | Ours($\lambda = 5 \times 10^4$) | Ours($\lambda = 1 \times 10^4$) |

Figure 14: Qualitative comparison of our removal attack on the Tree-Ring watermarking method.

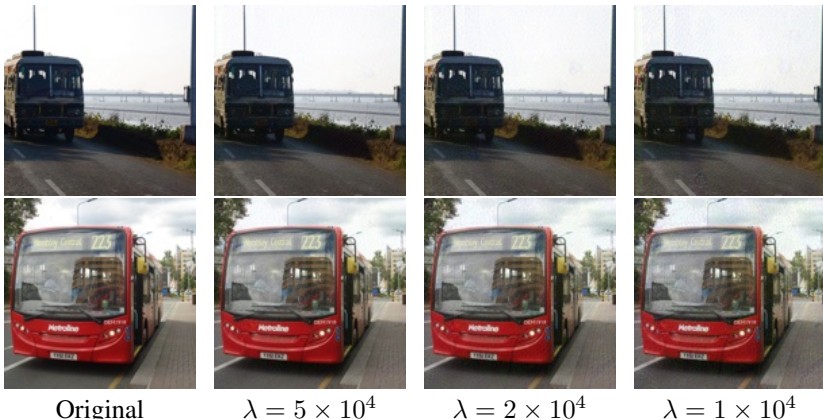

| Original | $\lambda = 5 \times 10^4$ | $\lambda = 2 \times 10^4$ | $\lambda = 1 \times 10^4$ |

Figure 15: Examples showing successful watermark forgery attacks on the Tree-Ring watermarking method with different hyperparameter $\lambda$ values.

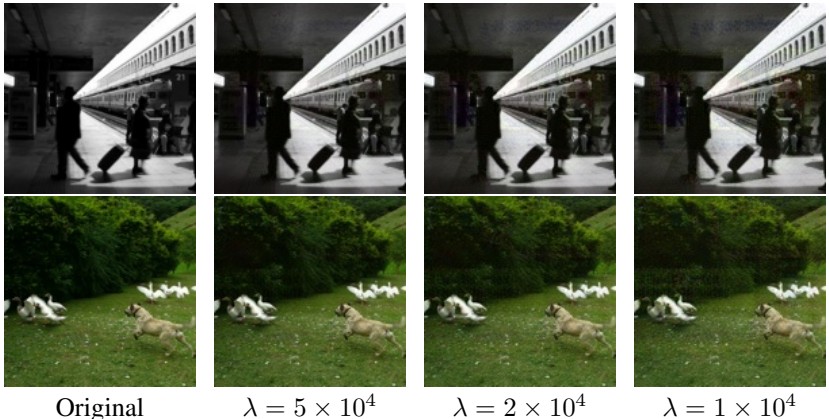

Original   $\lambda = 5 \times 10^4$   $\lambda = 2 \times 10^4$   $\lambda = 1 \times 10^4$

Figure 16: Examples showing successful watermark forgery attacks on the RingID watermarking method with different hyperparameter $\lambda$ values.

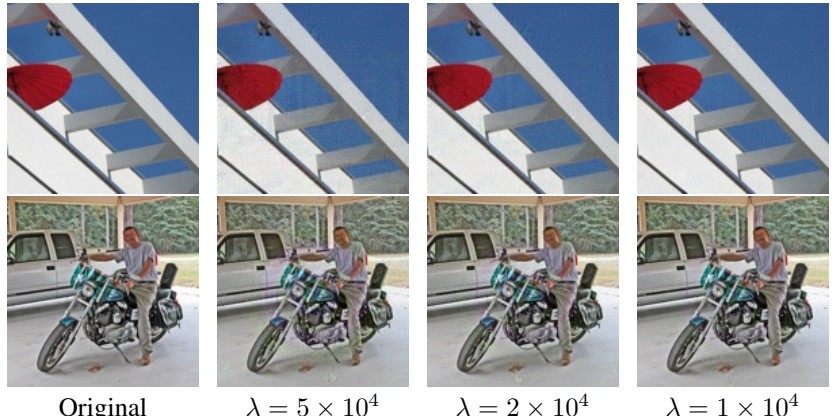

Original   $\lambda = 5 \times 10^4$   $\lambda = 2 \times 10^4$   $\lambda = 1 \times 10^4$

Figure 17: Examples showing successful watermark forgery attacks on the WIND watermarking method with different hyperparameter $\lambda$ values.

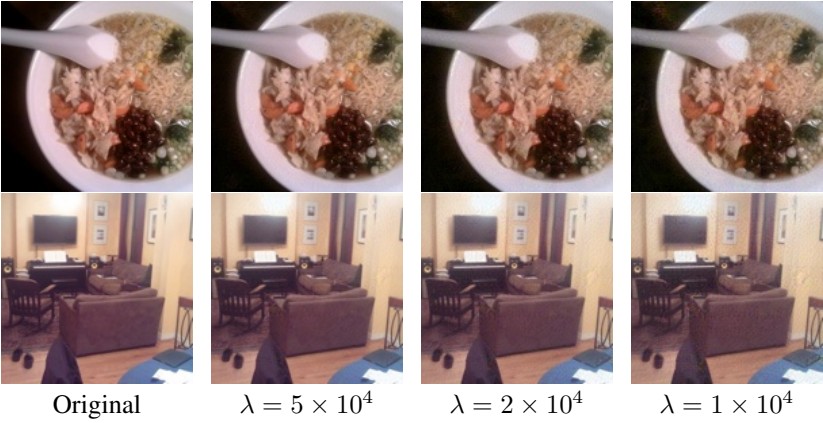

Original   $\lambda = 5 \times 10^4$   $\lambda = 2 \times 10^4$   $\lambda = 1 \times 10^4$

Figure 18: Examples showing successful watermark forgery attacks on the Gaussian Shading watermarking method with different hyperparameter $\lambda$ values.

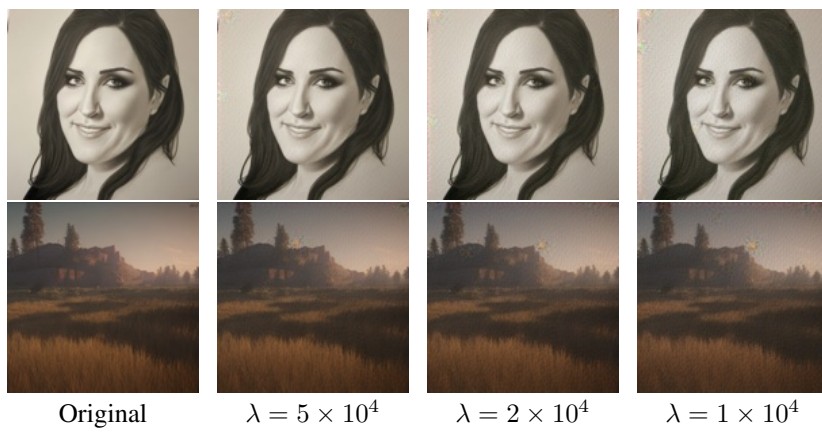

| Original | $\lambda = 5 \times 10^4$ | $\lambda = 2 \times 10^4$ | $\lambda = 1 \times 10^4$ |

Figure 19: Examples showing successful watermark removal attacks on the Gaussian Shading watermarking method with hyperparameter $\lambda$ values.

