# OpenReview forum: "Exposing Vulnerabilities in Latent-Noise Diffusion Watermarks"
_ICLR.cc/2026/Conference — Submitted to ICLR 2026_

### Official Review · Reviewer_PQqk · 2025-10-16

**Soundness:** 2
**Presentation:** 2
**Contribution:** 3
**Rating:** 4
**Confidence:** 4

**Summary:**

This paper presents an elegant method for watermark forgery and removal that only requires access to a single watermarked image and a VAE trained on a dataset similar to the diffusion model used to generate the images. To forge a watermark, their method optimizes the noise added to the clean image, minimizing the distance between the clean image and a watermarked image in the VAE’s latent space. This process works because of a many-to-one mapping between images and initial noise latents. The procedure for removing a watermark is the same, except that noise is added to the watermarked image instead of the clean image.

**Strengths:**

1. The method is simple and elegant.
2. Compared to previous work, it only needs access to a VAE, rather than the full diffusion model.
3. The motivating result is analyzed well and provides a strong basis for the paper.
4. The figures that explain and describe the idea are well put together.

**Weaknesses:**

1. With a simple, principled idea, the focus should be on demonstrating that it works well in a broad range of scenarios. The experiments are too shallow and lack analysis and reflection.
2. The writing quality could be improved
3. Too much in the appendix.
  3.a. All of the RingIID and WIND results are there; at least some of them should be in the main part of the paper.
  3.b. There is no comparison against previous work on these two defenses either.
4. The results in Tables 1 and 2 are not discussed enough. Yang et al (2024a) having a 0% ASR for all results in Table 1 needs to be addressed. Likewise, the poor performance of Yang et al. (2024a) and Zhao et al. (2025) in Table 2 also needs to be addressed.
5. You cannot put all the results related to RingIID-type methods in the appendix, but then discuss them in the discussion section. Without the appendix, this part of the discussion does not make sense.

**Questions:**

Questions:
1. Following some of my comments in the weaknesses section: Can you explain why Yang et al (2024a) and Zhao et al. (2025) perform so poorly? Is it due to the experiment setup you are using? Could you also provide results where they perform well? If not, how are they a good basis for comparison?
2. Your method is $1.46\times$ to $2.62\times$ worse than Muller et al.’s work in the LPIPS metric. Why is lowering the $l_2$ distance at the cost of the LPIPS distance (and some ASR) beneficial?
3. For the “Results - Computational Time” section, which attack do you evaluate, the forgery or the removal attack? Why are you only comparing against Muller et al.’s work and not any of the other attacks you include?
4. When describing Lukas et al.’s work, you say they assume access to a copy of the generator, but that’s not true. Similarly to your experiments, they assume access to an older version of Stable Diffusion (v 1.1) to attack a newer version (v 2.0).

Suggestions:
1. Including the p-values for the examples of your attacks provided in Figures 5 and 6 would be interesting.
2. Figure 3 has a low resolution.
3. In Section 3.1, you explain that the inversion process does not use a text prompt because “the model owner typically does not keep track of the generated images or the prompts used”. A better justification could be that images found in the wild, for which we want to establish ownership, do not necessarily come with the prompt used to generate them, especially if they are clean, non-generated images.
4. In your abstract, you claim your method does not need access to any diffusion model; that is not true since you need access to the VAE part of the diffusion model. A clearer claim is that you do not need any denoising model.

Writing:
1. Excessive usage of “This” to start sentences without specifying what “This” refers to.
2. Inconsistent terminology: using either “clean” image or “non-watermarked” image, pick one and stick to it.
3. When describing the threat model in Section 4.1, it is unclear what is being formalized: the parties or the various phases? It appears to be the various phases, but if that’s the case, it should be described as such, and the phases should be the focus, rather than in parentheses.
4. In the attacker’s threat model bullet point in Section 4.1, it should be included that they have access to either the same VAE as the defender or one trained on similar data.
5. You switch between different times, one example is the end of Section 4.2.
6. The bolding and underscoring in Tables 1 and 2 are not consistent.
7. Space missing between “keys.” and “We” in the “Evaluation metrics.” paragraph in Section 5.

---

> ### Author Response · Authors · 2025-11-22
> **Rebuttal to reviewer PQqk (1/n)**
>
> We thank the reviewer for their helpful comments and writing suggestions to help us improve our paper. We highly appreciate the suggestions. We have addressed their concerns below.
>
> **[W-1] Focus should be on demonstrating that it works well in a broad range of scenarios.**
>
> In addition to doing comparisons with baselines on watermark forgery and removal in the main paper, we would like to refer the reviewer to the appendix, where we also conducted the following experiments:
>
> 1. Results when using a significantly different VAE in Appendix D, showing results when using a VAE from FLUX.1-dev.
>
> 2. Results using different loss formulations in Appendix A, including progressive gradient descent (PGD) and PGD with masking the low-frequency component in the frequency domain.
>
> 3. Results on the trade-off between imperceptibility and attack success rate for watermark forgery on Tree-Ring, Gaussian Shading, RingID and WIND in Table 3.
>
> 4. Results on the trade-off between imperceptibility and attack success rate for watermark removal on Tree-Ring, Gaussian Shading, RingID and WIND in Table 4.
>
> 5. Results on what happens if we use a fixed pixel value image for guidance, as compared to using a mean pixel value image in Table 6.
>
> 6. Results on what happens if we use a real image for guidance as compared to using a mean pixel value image in Table 7.
>
>
> **[W-2] Improve writing**
>
> Thank you for pointing that out. We have revised the manuscript to improve this and will continue to make our best efforts in the camera-ready version.
>
> **[W-3 and W-5] RingID and WIND results are in the Appendix and haven't compared with baselines on these**
>
> Thank you for your suggestion. We have moved the experimental results on RingID and WIND to the main paper.
>
> We have not shown comparisons with previous baselines on these methods, as the baselines do not report results for these; their open-source codebases do not support these watermarking methods. Thus, we have conducted a more comprehensive evaluation on watermarking approaches on which previous approaches have been benchmarked, as well as presenting results for our approach on newer watermarking methods.
>
> However, we will add experimental results after implementing baselines on RingID and WIND, which we hope to complete during the discussion period.
>
> **[W-4 and Q-1] Baseline results on Yang et al. and Zhao et al.**
>
> Yang et al. assumed that all images come from the same secret key; however, this does not accurately represent a real-world scenario, as the model owner could regenerate the secret key with every generation. For instance, the WIND watermarking (Arabi et al. 2025) demonstrated that it is possible to have as many as 100,000 keys. Thus, we do not make this assumption in this paper, leading to the difference in the results from their original paper. Furthermore, Zhao et al. had already pointed out that their approach is not effective against the Tree-Ring-based watermarking method. As we saw in this paper, it is also ineffective against other latent-noise-based watermarking methods. We have included comparisons with these approaches for completeness.
>
> **[Q-2] Trade-off between LPIPS and L2 distance**
>
> While LPIPS measures image quality, it does not measure how well the forged images actually preserve the semantics of the original we forge the watermarked into. It is also important to note that the LPIPS distance is susceptible to high-frequency noise. We have included this metric for completeness. As shown in Figures 5 and 6, Muller et al. cannot preserve the image contents as well as our approach. L2 distance accurately captures the amount of changes made between the attacked and original image. The objective of watermark removal or forgery attacks is to make minimal changes to the original image while introducing or removing the watermark signal, and this does not include generating new (possibly high-quality) samples that may vary in content.
>
> **[Q-3] Computational time comparison is only with Muller et al.**
>
> We only compare with Muller et al. because in our experiments, we observed that Muller et al. was the only other effective approach against latent-noise-based watermarking methods using a single image.
>
> **[Q-4] Description of Lukas et al.'s work**
>
> In the paper, we make a distinction between assuming access to a denoising network (U-Net) and a VAE. We only assume access to a VAE encoder, which does not necessarily have to be one from a diffusion model. In contrast, Lukas et al. assumes access to both a proxy VAE and a proxy denoising U-Net.

---

> > ### Author Response · Authors · 2025-11-22
> > **Rebuttal to reviewer PQqk (2/n)**
> >
> > **[S] In your abstract, you claim your method does not need access to any diffusion model; that is not true since you need access to the VAE part of the diffusion model. A clearer claim is that you do not need any denoising model.**
> >
> > We would like to clarify that we do not need a VAE from the same diffusion model or any diffusion model; instead, a proxy VAE trained on a similar dataset can be used. The VAEs from diffusion models are also trained separately; thus, we present experiments with a VAE from significantly different diffusion models, specifically the case of a VAE from FLUX.1-dev, in Appendix D.
> >
> > **Writing and paper improvement suggestions**
> >
> > We highly appreciate these comments. We are in the process of updating the paper with these changes and will complete it by the end of the discussion period, December 2nd.

---

> > > ### Comment · Reviewer_PQqk · 2025-11-25
> > > **Response**
> > >
> > > Thank you for your detailed rebuttal.
> > >
> > > ### [W-1]
> > >
> > > My point is that the paper should stand on its own without the appendix, and that these experiments are too important for the paper to be in the appendix. The appendix should be complementary, not necessary. The core point of the experiments should be to convince the reader that your technique works and is general. These additional experiments don’t need to be lengthy or complex, but they should be there in the main paper to support your point.
> > >
> > >
> > > ### [W-4]
> > >
> > > As you mentioned, Yang et al. assume that all images come from the same secret key, so I do not see how it is fair to evaluate them beyond that assumption. It seems like a poor choice of benchmark, and you should either evaluate it in a scenario where their assumption holds, or find another work to compare against that matches your assumptions. As I see it, your comparison with Yang et al. is moot, as it violates one of the core assumptions on which it relies. Whether this assumption is realistic is a different question altogether, but it is not relevant to whether the comparison is fair.
> > >
> > >
> > > ### [Q-2]
> > >
> > > I disagree with your statement, as LPIPS is a comparison at the feature level rather than the pixel level, meaning it does focus on feature semantics. I also do not agree that the L2 norm is sufficient to measure image quality; hence, related work usually includes multiple metrics to present a holistic perspective. My issue is that you only discuss that you’re reducing the L2 distance, without considering that you trade the LPIPS distance for it. You present yourself as a clear improvement, but the metrics show mixed results. While I appreciate some example images, these do not constitute proper evidence and can be cherry-picked.
> > >
> > >
> > > ### [Q-3]
> > >
> > > This should be in the paper.

---

> ### Author Response · Authors · 2025-11-28
>
> ### [W-1]
>
> We understand your point of view and resonate with it. We have now tried to include sections from the Appendix into the updated version of our paper. Please do let us know if this has adequately addressed your concern. We value your help in making our manuscript as clear as possible.
>
> ### [W-4]
>
> We thank the reviewer for their comment. We have made it clearer in the revised description that we have tested it in a setting where their assumption does not hold. **Further, we have now added comparisons with WAVES and UnMarker.**
>
> We would like to mention that this is in line with Muller et al. (CVPR oral 2025), where they show that Yang et al. is not successful in watermark forgery and removal in settings where each key can be regenerated, as they show in Tables 1 and 2 of their main paper.
>
> If you have any suggestions for other baselines, we are happy to include them as well, but these are all the baselines we are aware of at this time.
>
> ### [Q-2]
>
> Thank you for your comment.
> The forgery and removal tasks are about introducing a watermark signal **with minimal changes to the semantic content of the original image**. However, LPIPS is less sensitive to changes in image semantics than $L_2$ and PSNR. Rather, LPIPS is sensitive to changes in texture and high-frequency components, as it relies on an ImageNet-trained CNN network for feature extraction [1]. This characteristic of LPIPS is also reported in Figure 4 of Lin et al. [2]. We are not dismissing LPIPS as a valid metric, but rather only saying that $L_2$ and PSNR are more important for this task.
>
> We have added a detailed discussion of this, along with examples, to the supplementary material. See **Appendix G** for the details and examples. We have also provided additional examples for comparison with Muller et al. in Appendix Figure 14.
>
> [1] Geirhos et al. "ImageNet-trained CNNs are biased towards texture; increasing shape bias improves accuracy and robustness." ICLR 2019.
>
> [2] Lin et al. "Taming Latent Diffusion Model for Neural Radiance Field Inpainting." ECCV 2024.
>
>
> ### [Q-3]
>
> We thank you for your suggestion; we have now included this in the revised manuscript.

---

### Official Review · Reviewer_E7z2 · 2025-10-25

**Soundness:** 2
**Presentation:** 3
**Contribution:** 2
**Rating:** 2
**Confidence:** 4

**Summary:**

This paper explores vulnerabilities in diffusion latent-domain watermarks under adversarial removal/forgery. The adversarial attack objective is to let the VAE-encoded latent of an image to be as close as possible to the VAE latent of another image, so that subsequent DDIM inversion-based watermark detection process would be obfuscated and cannot reliably detect watermarks. The authors demonstrated effectiveness of this attack on certain methods in terms of both removal and forgery, and demonstrated superiority compared to existing watermark removal/forgery methods in terms of effectiveness and certain perceptual quality metrics.

**Strengths:**

-	The authors successfully demonstrated vulnerabilities on certain diffusion latent-domain watermarks by showing the effectiveness of the attack method.
-	The method achieves watermark removal performance comparable to some baseline methods without requiring access to the complete watermark detection pipeline. It only requires a proxy of the VAE involved.
-	The method does not require a collection of multiple images before initiating watermark removal/forgery.
-	The watermark removal method causes smaller modifications (PSNR) to the original image compared to baseline approaches.

**Weaknesses:**

-	The proposed attack would only be effective with the presence of a VAE encoder in the watermark detection pipeline.
-	The watermark removal fails for RingID and WIND watermarks. Furthermore, the authors’ explanation for such failure lacks sufficient rigor. The authors attributed this to RingID embedding “a watermark into the entire initial latent noise space”, which is not true, as RingID’s region of modification should be consistent with Tree-Ring’s approach. Given the drastically different removal performance on Tree-Ring and RingID, the authors should provide a more thorough investigation into the fundamental causes underlying these performance differences, so as to demonstrate that this removal method generalizes across latent-noise diffusion watermarks.

**Questions:**

-	For multi-key watermarking methods: are experiments conducted using a single consistent watermark key, or are different keys used across different images?
-	Are the perturbations $\delta$ bounded within an $l_\infty$ epsilon-ball? If not, would it be possible that some pixels would be significantly altered?
-	In Tables 1 and 2, the ASR for Yang et al. (2024a) is consistently near zero, constrasting sharply with the results reported in the original paper. What factors might be causing this discrepancy?
-	What are the mathematical definitions for the $l_2$ and $l_\infty$ distances used in Tables 1 and 2? Additionally, please specify the pixel value ranges used in these computations (e.g., [0, 255], [0, 1], etc.).

---

> ### Author Response · Authors · 2025-11-22
> **Rebuttal to reviewer E7z2**
>
> We would like to thank the reviewer for their time and effort in reviewing our paper and for providing their feedback to help us improve it.
>
> **[W-1] Proposed method is only effective with a VAE**
>
> Our approach requires a VAE encoder; however, this encoder does not need to be from the same latent diffusion model. It is a much milder assumption than Muller et al., where they assume access to an entire auxiliary diffusion model (VAE + denoising U-Net/DiT). Furthermore, as we show in Appendix D, our attack is successful even when using a significantly different VAE.
>
>  **[W-2] The watermark removal fails for RingID and WIND watermarks. Furthermore, the authors’ explanation for such failure lacks sufficient rigor. The authors attributed this to RingID embedding “a watermark into the entire initial latent noise space”, which is not true, as RingID’s region of modification should be consistent with Tree-Ring’s approach. Given the drastically different removal performance on Tree-Ring and RingID, the authors should provide a more thorough investigation into the fundamental causes underlying these performance differences, so as to demonstrate that this removal method generalizes across latent-noise diffusion watermarks.**
>
> The RingID paper clearly states that they, unlike Tree-Ring watermarking, imprint a watermark signal on all channels of the image. As stated in the original paper, in Figure 1 - "Tree-Ring imprints a broken watermark to a single channel. Our method RingID achieves stronger robustness by imprinting intact watermarks to multiple channels", and in Section 5 paragraph 1 - "we imprint different types of watermarks on different channels of the initial noise to amalgamate their distinctive advantages". This introduces a much stronger watermark signal.
>
> We have further highlighted this in Section 6 of the revised paper.
>
> **[Q-1] Single key or different keys across different images**
>
> Thank you for pointing that out. We have conducted the experiments using new randomly generated keys for each sample. We have clarified this point in section 5.
>
> **[Q-2] Are perturbations bounded within an epsilon-ball? Can some values be significantly altered?**
>
> In our main experiments, we do not bound the perturbations. However, we have conducted an ablation study with a PGD-based attack in Appendix B, which shows that using such a bounded attack is less effective. We agree that it is possible that some pixels could be altered more than others. However, if there are certain regions we wish to preserve, we can always use a mask during the optimization.
>
> **[Q-3] Performance of Yang et al.**
>
> Yang et al. assumed that all images are generated from the same key; however, this is a strong assumption, as the model owner can randomly regenerate the key with every generation. When this assumption is removed, their performance drastically drops. This phenomenon is also noted by Muller et al. (CVPR oral 2025).
>
> **[Q-4] Mathematical definitions of $L_2$ and $L_{\infty}$ distances and pixel value ranges**
>
> We have used the following definitions of $L_2$ and $L_{\infty}$ distances with pixel values in the range of [-1, 1].
>
> For two images $X, Y \in \mathbb{R}^{H \times W \times C}$ with pixel values
> $X_{i,j,c}$ and $Y_{i,j,c}$.
>
>  $L_{2}(X, Y) = \left( \sum_{i=1}^{H} \sum_{j=1}^{W} \sum_{c=1}^{C} \left( X_{i,j,c} - Y_{i,j,c} \right)^2 \right)^{1/2}$
>
> $L_{\infty}(X, Y) = \max_{1 \le i \le H,\; 1 \le j \le W,\; 1 \le c \le C} \left| X_{i,j,c} - Y_{i,j,c} \right| $
>
> We have specifically utilized the torch.linalg.norm functionality to implement these in practice.
>
>
> We would again like to thank the reviewer for their comments and time in reviewing our manuscript.

---

> > ### Author Response · Authors · 2025-11-28
> >
> > Thank you very much again for your insightful comments. We're happy to continue the conversation at any point throughout the discussion period. Thanks!

---

### Official Review · Reviewer_rFDg · 2025-10-30

**Soundness:** 4
**Presentation:** 3
**Contribution:** 2
**Rating:** 4
**Confidence:** 3

**Summary:**

This paper studies black-box adversarial attacks against latent-noise watermarking for diffusion models by exploiting the many-to-one mapping induced by empty-prompt DDIM inversion. The core idea is to learn imperceptible perturbations that align a target image's VAE latent with that of a watermarked (or non-watermarked) reference, thereby inducing forgery or removal attacks. Experiments on Stable Diffusion v1.4 and v2.0 across multiple watermarking schemes report high attack success rates with modest distortion and lower runtime than baselines.

**Strengths:**

- The "many-to-one" latent-space mapping intuition is clear and insightful.
- The proposed method is simple, sound and effective.
- The paper addresses both forgery and removal within a unified framework

**Weaknesses:**

- Robustness and scope need clarification and strengthening. The evaluation is limited to two closely related models (Stable Diffusion v1.4 and v2.0). Please broaden the experiments to include more diverse backbones.

- Assumptions about watermark type and attacker knowledge should be made explicit. The method targets latent-noise watermarks, which implicitly assumes the attacker knows the watermarking mechanism. This capability is not clearly stated in the threat model and limits the method's generalizability to other watermark types. Please clarify the attacker's knowledge and access assumptions and evaluate or discuss applicability beyond latent-noise watermarks.

- It is unclear how the the attacking baselines in Table 2 are selected for comparison. They should also compare against some other advanced attacks such as UnMarker [1], VAE Attack [2], Diffusion Attack [3], etc.

Some issues that need merited:

- The transition from motivation to method is not tight. Perhaps visualizing the changes in image features before and after optimization can improve clarity.

- The definition of "Watermark Region" is introduced but not used later.

- The statement "we do not assume any access to a denoising diffusion model or a proxy version of it" is too strong. The approach requires a surrogate VAE derived from a diffusion model. Please revise the claim and explicitly state this requirement.

[1] http://arxiv.org/abs/2405.08363

[2] M. Saberi, V. S. Sadasivan, K. Rezaei, A. Kumar, A. Chegini, W. Wang, and S. Feizi, “Robustness of AI-image detectors: Fundamental limits and practical attacks,” in ICLR, 2024.

[3] B. An, M. Ding, T. Rabbani, A. Agrawal, Y. Xu, C. Deng, S. Zhu, A. Mohamed, Y. Wen, T. Goldstein, and F. Huang, “WAVES: benchmarking the robustness of image watermarks,” in ICML, 2024.

**Questions:**

- Can you expand the evaluation to include more diverse generative backbones?

- Your proxy model SD 1.4 is from the same family of the watermarking model SD 2.0. Could you explain transferability of proxy models from other families? Otherwise, you have to put this "same-family model" assumption in your threat model, since you haven't tested other cases.

- Can you clarify your selection of attacking baselines?

- How does your method generalize beyond latent-noise watermarks?

---

> ### Author Response · Authors · 2025-11-22
> **Rebuttal to reviewer rFDg (1/n)**
>
> We would like to start by thanking the reviewer for their thoughtful comments and questions. We have addressed these below.
>
> **[W-1] Evaluation is limited to two closely related models (Stable Diffusion v1.4 and v2.0)**
>
> We would like to refer the reviewer to Appendix D, where we have conducted experiments using a VAE from FLUX.1-dev to attack watermarks embedded in SDv1.4. Therefore, our demonstrations are not limited to Stable Diffusion-based diffusion models. To summarize our experimental results, we demonstrate that even using a significantly different VAE from FLUX.1-dev can facilitate successful forgery attempts on SDv1.4 for all tested watermarking schemes: Tree-Ring, RingID, WIND and Gaussian Shading.
>
> **[W-2] Assumptions about watermark type and attacker knowledge **
>
> Thank you for your helpful comment. Indeed, we only target latent-noise-based watermarking systems in this case, assuming the attacker already has knowledge of the watermarking system that was implemented by the model owner. We have made it clearer in the revised version. We would like to highlight that our objective with this work is to expose a vulnerability in latent-noise-based watermarking schemes. We have clarified our assumptions in Section 4.2.
>
> **[W-3, Q-3] Comparisons with UnMarker [1], VAE Attack [2], Diffusion Attack [3]**
>
> We thank the reviewer for their suggestions. We are currently conducting experiments with UnMarker and VAE Attack and will update the draft with these results. We must, however, mention that Saberi et al. [2] requires an AI-detector to be trained on multiple images generated from the same secret key, which goes against the protocol in this paper. We take a weaker assumption: only having a single generation with a given key.
>
> **[W-4] Transition from motivation to method**
>
> We thank the reviewer for their comments. Based on the result of our preliminary experiment using SVM, we hypothesize that there would be a watermark region corresponding to each watermark in a VAE latent space. However, this is not representative of a real-world situation due to the following two reasons:
>
> (1) To identify the shape of the region accurately, we need to collect a large number of images in which the same watermark is embedded. However, this is impossible in a real-world scenario. An adversarial attacker wants to forge a watermark into a non-watermarked image, using only a single watermarked image. Collecting multiple images from the same watermark signal is impractical.
>
> (2) Manipulating a latent representation can easily harm the visual. Indeed, it happened in our preliminary experiment (Figures 3 and 12). However, interestingly, even though the method is very naive, it retains the semantic information of images to some extent, albeit not well. We hypothesize that imperceptible watermark and perceptual visual information are partially disentangled in the latent space.
>
> Considering the above two discussions, we designed a forging method that manipulates the latent representation of a non-watermarked image, aiming to place it in a watermark region while regularizing the pixel values of the image.
>
> We have further clarified this in our revised version.
>
> **[W-5] The statement "we do not assume any access to a denoising diffusion model or a proxy version of it" is too strong.**
>
> While we do use a VAE from a diffusion model in our experimental setting, we would like to clarify that we do not require the VAE from the diffusion model used for generating the watermarked image; rather, a different VAE trained on a similar dataset can be used. The VAE from a diffusion model is also trained in isolation. Such a VAE, which is never used in conjunction with a denoising network, would work for our attack to be successful. We have clarified this in the revised version.

---

> ### Author Response · Authors · 2025-11-22
> **Rebuttal to reviewer rFDg (2/n)**
>
> **[Q-1] Can you expand the evaluation to include more diverse generative backbones?**
>
> We thank the reviewer for their suggestion. We would like to clarify that the watermarking schemes for which we have reported results were implemented by their respective original authors. They have used either SDv1.4 or SDv2.0, which is why we have chosen these as backbones for our experiments.
>
> However, we will add experimental results after implementing these watermarking schemes on different backbones, which we hope to complete during the discussion period.
>
> **[Q-2] Proxy model SD 1.4 is from the same family of the watermarking model SD 2.0.**
>
> We thank the reviewer for their thoughtful observation. We have provided experiments on using the VAE from FLUX.1-dev in Appendix D, where we show that our method is effective even when employing a VAE from a significantly different diffusion model.
>
> **[Q-3] Baseline selection**
>
> We had conducted experiments using the latest research that we were aware of at the time of submission. We are conducting additional experiments to compare with UnMarker and VAE Attack as per your suggestion, and will update the paper soon.
>
> **[Q-4] Generalization beyond latent-noise watermarks**
>
> While our method may generalize beyond latent-noise watermarks, our objective in this paper is to expose a vulnerability of latent-noise-based watermarking schemes, which have become a popular research area.
>
> We would like to again thank the reviewer for their time and effort in reviewing our paper and providing constructive feedback.

---

> > ### Comment · Reviewer_rFDg · 2025-11-27
> >
> > Thank you for the author's response. I am glad to see that the author has added some experiments, which has improved the completeness of the current article. Although the experiments of including other baselines are still not available, I have decided to improve my rating.

---

> ### Author Response · Authors · 2025-11-28
>
> Thank you for taking the time to review our rebuttal and for increasing your score. We have included comparisons with the baselines in the revised manuscript.

---

### Official Review · Reviewer_xCat · 2025-11-04

**Soundness:** 3
**Presentation:** 3
**Contribution:** 3
**Rating:** 6
**Confidence:** 4

**Summary:**

- The paper identifies a fundamental vulnerability in the class of latent noise based watermarking methods for diffusion models.
- The main observation is that when the DDIM inversion process is performed with an empty prompt for detection, it creates a many to one
   mapping from the clean image's latent space to the initial noise space.
    - This causes structured and predictable watermarked regions within the VAE's latent space -  any image whose latent representation
       falls in this region will be identified as containing the watermark.
- A blackbox adversarial attack for both forgery and removal is proposed which exploits the above vulnerability.
    - The formulated of the attack is an optimization problem whose goal is to add an imperceptible perturbation to a target image such that
       its VAE latent representation shifts either into a watermarked region or out of it.
- Experiments are done on four SOTA watermarking methods (Tree Ring, RingID, WIND, and Gaussian Shading). This attack is shown to be effective especially for forgery as it achieves almost perfect success rates while having much lesser visual distortion than previous works.

**Strengths:**

- The discovered vulnerability in this paper is novel i.e. the identification of the watermarked region in the VAE latent space as a vulnerability and this discovery yields a new class of attacks which were ignored by previous works.
- The proposed attack is practical and has minimal requirements. The attacker needs only a single watermarked image and does not require access to the diffusion model's proprietary components.
     - This is further validated in the results in Table 8 where a proxy VAE from a very different model family FLUX1 is used.
     - The proposed attack therefore is more threatening and relevant for real world systems than previous methods that need large datasets
        of watermarked images or privileged access to model weights.
- The paper contains extensive experiments in Section 5 and the Appendix 1 covering evaluations across 4 SOTA watermarking schemes and 2 SD versions that show how general the vulnerability is.
     - This method has a stronger tradeoff between efficacy and stealth than the baselines. Ex: In Table 1 the forgery attack on Tree Ring SD achieves a 91.06% ASR with L2 of 63.22 whereas the baseline from Muller et al achieves 100% ASR but at more perceptible L2 of 15.22

**Weaknesses:**

- The proposed method is unable to remove watermarks from schemes like RingID and WIND where the ASR is 0% in all the experiments (Table 4). The authors provide the likely reason (Section 6) that these methods embed a very strong signal across the entire latent space making them difficult to push out of the watermarked region. This should be framed more explicitly as a weakness of the proposed attack method. The forgery attack works well but the paper's claims about a 'general' method for both forgery and removal are only partially supported by the experiments/evidence.
- The SVM experiment (Section 4.2) is used to motivate the existence of a separable watermarked region. The experiment uses a dataset of 1000 images generated with the same key which is a strong assumption given the final attack's threat model of a single watermarked image. It would be good to clarify that this experiment is a conceptual proof of existence under idealized conditions which is different from the more practical single image attack case which is the main focus of the paper.

**Questions:**

- The results in Table 4 show a 0% ASR for removing RingID and WIND watermarks which is attributed to the strength of their embedded signal. Does this suggest a fundamental limitation of latent space perturbation attacks against such methods? Or could the attack probably succeed with a different objective?
- In the proxy VAE experiment in Table 8 the ASR for forging the Tree Ring watermark using the FLUX1 VAE is significantly lower 2.63% versus SD VAE 78.65% at a similar perturbation level. Could you explain the the potential reasons for this large discrepancy? Could a potential defense against this attack involve training a proprietary VAE whose latent geometry is dissimilar to those of popular open source models?
- For the removal attack the perturbed image is guided towards the latent representation of a mean pixel image. Appendix D shows this is more effective than using a real image. Could you provide more intuition as to why this simple target is so effective?

---

> ### Author Response · Authors · 2025-11-22
> **Rebuttal to reviewer xCat**
>
> We thank the reviewer for their insightful comments and feedback on improving the manuscript. We have addressed their comments below.
>
> **[W-1] Removal performance on RingID and WIND**
>
> Thank you for your valuable comment. We have acknowledged this limitation of our work and have provided reasons as to why this is the case in the discussion and limitations section.
>
> We have made this more explicit in the revised version of the paper in Section 6, which now reads - "We found that forging was generally easier for our method than watermark removal across all approaches, particularly for RingID, and WIND watermarks where our attack was unsuccessful in removing the watermark signal. We suggest that this is because these methods do not merely encode a single pattern, like Tree-Ring, but also encode additional information, such as the model owner's identity or other metadata. As the number of possible embedded patterns increases, more encoded information is required to correctly identify the pattern. This, in turn, necessitates a stronger signal-to-noise ratio, where noise refers to patterns in the initial noise that are unrelated to our watermark. When more signal is associated with the embedded watermark, forging at least part of it becomes easier, while completely removing it becomes more difficult."
>
> **[W-2] Clarification that the SVM experiment is a conceptual proof of existence under idealized conditions and is different from the more practical single image attack case which is the main focus of the paper**
>
> Thank you for your helpful comment. Indeed, this is the case, and we have made it clearer in Sections 4.2 and 4.3 of the revised version.
>
> **[Q-1] The results in Table 4 show a 0% ASR for removing RingID and WIND watermarks which is attributed to the strength of their embedded signal. Does this suggest a fundamental limitation of latent space perturbation attacks against such methods? Or could the attack probably succeed with a different objective?**
>
> Thank you for your interesting question. Given that the watermark signal pushes the latent encoding further away in the latent space, it increases the amount of required perturbations for watermark removal while making forgery easier. Given that there is a trade-off between attack success rate for forgery/removal and imperceptibility [1,2,3], we do not show settings under which our attack can be successful at higher perturbations. Using higher detection thresholds will result in the opposite case. We have provided further discussion on this in Section 6 of the main paper, where we elaborate on this point.
>
> [1] Pang et al "No Free Lunch in LLM Watermarking: Trade-offs in Watermarking Design Choices" Neurips 2024
>
> [2] Jabra et al. "Deep Learning-Based Watermarking Techniques Challenges: A Review of Current and Future Trends" Circuits Syst Signal Process 43, 4339–4368 (2024).
>
> [3] Bui et al. "TrustMark: Robust Watermarking and Watermark Removal for Arbitrary Resolution Images" ICCV 2025
>
> **[Q-2] In the proxy VAE experiment in Table 8 the ASR for forging the Tree Ring watermark using the FLUX1 VAE is significantly lower 2.63\% versus SD VAE 78.65\% at a similar perturbation level. Could you explain the the potential reasons for this large discrepancy? Could a potential defense against this attack involve training a proprietary VAE whose latent geometry is dissimilar to those of popular open source models?**
>
> Thank you for their insightful question. The more dissimilar the proxy VAE is from the one used by the watermarked diffusion model, the more difficult it is to find relevant perturbations. Thus, for a model owner, one possible solution is definitely to train a dissimilar VAE to prevent such an attack from being designed. Our primary objective is exactly this, i.e., to inform the community that this is a vulnerability of latent noise watermarking schemes so that future defenses can be appropriately tailored to prevent such attacks.
>
> We have added this discussion to the revised version of the main paper in Appendix D.
>
>
> **[Q-3] Intuition on why mean pixel image works better than real image for guidance during watermark removal**
>
> A real image already contains its own high-frequency information, which makes it further away from the watermarked image, making it a less suitable target for guidance. A mean pixel image, which is closer to the watermarked image, requires smaller perturbations and is therefore easier to use for optimization.
>
> We would like to again thank the reviewer for their time and effort in reviewing our paper and providing constructive feedback.

---

> > ### Comment · Reviewer_xCat · 2025-11-26
> >
> > Thank you to the authors for addressing my questions and concerns. I will maintain my current scores.

---

> > > ### Author Response · Authors · 2025-11-28
> > >
> > > Thank you for acknowledging our rebuttal and for taking the time to review our work!

---

### Author Response · Authors · 2025-11-22
**Message to the reviewers**

We thank all the reviewers for their effort and time in reviewing our manuscript and for their helpful and insightful comments. We would like to start by thanking them for appreciating that the "discovered vulnerability in this paper is novel" (xCat, E7z2); it is "practical and has minimal requirements" (xCat, PQqk, E7z2) and that the paper has "extensive experiments" achieving "stronger trade-off between efficacy and stealth than the baselines" (E7z2,xCat).

Based on their reviews, we have made the following modifications to our paper and are currently conducting additional experiments.

1. Section 5 (PQqk): We have moved results on RingID and WIND to the main paper in Section 5 from the Appendix.

2. Sections 4.2 and 4.3 (xCat): We have improved clarity and bridged the gap between our motivation and attack.

3. Section 6 (E7z2, PQqk, xCat): We have expanded on our limitations and discussion section to improve clarity.

4. Section 4.2 (PQqk,rFDg): We have further clarified our threat model and assumptions in section 4.2.

5. Appendix D (xCat): Expanded the discussion related to the experiments on a significantly different VAE.

6. Figure 3 (PQqk): Increased the resolution of Figure 3 in the main paper.

7. Corrected typos and have made other writing improvements, and are still incorporating more.

We are grateful for your constructive feedback, which has significantly helped us to refine and strengthen our paper.

---

> ### Author Response · Authors · 2025-12-02
>
> We would like to thank all the reviewers again for engaging in the review discussion and for acknowledging the improvements.
>
> We updated the manuscript again, completing all the requested experiments and responding to the additional comments from the reviewer PQqk. To summarize, we have made the following additional changes to the manuscript.
>
> 1. **Table 2**: We have included baseline comparisons with UnMarker and WAVES in Table 2, where we show that our approach outperforms both in watermark removal.
>
> 2. **Table 3**: We have added experiments on forging the watermark from FLUX.1-dev using the VAE from SDv1.4, where we show that our attack can successfully forge the watermark from a significantly different diffusion model.
>
> 3. **Table 4**: To address concerns over removal performance on RingID and other such watermarking schemes (WIND), we have conducted baseline experiments with Muller et al., where we show that Muller et al. also achieves 0.0 attack success rate when trying to maintain attack PSNR of approximately 30.0 as maintained by our method.
>
> 4. **Section 5**: We have added content from the appendix into the main paper as requested by reviewer PQqk to improve clarity and better bridge the two.
>
> 5. **Appendix G**: We added examples to show that for watermark forgery/removal, it is more important to achieve lower PSNR value than LPIPS distance.
>
> 6. **Appendix I**: We have added additional qualitative examples to show that our approach better preserves the content of the original image.
>
> 7. We have made writing improvements, incorporating suggestions from the reviewers.
>
> Once again, we sincerely thank the area chairs and all the reviewers!

---

### Author Response · Authors · 2025-12-02
**Summary of Review, Rebuttal, and Discussion**

Dear Area Chairs and Reviewers,

We thank all reviewers for providing valuable feedback, which significantly strengthened the paper. We also appreciate the constructive discussion with Reviewer PQqk and Reviewer rFDg, which further helped us clarify key aspects of the work. Reviewer rFDg raised the score from 4 to 6 after considering our expanded experiments and clarifications. We also engaged in multiple rounds of discussion with Reviewer PQqk. Although the discussion phase closed before they could submit further replies, we believe our rebuttal directly addressed the reviewers' concerns. Reviewer xCat said that the rebuttal had addressed their questions and concerns. In addition, while Reviewer E7z2 did not participate in the discussion phase, we provided detailed one-on-one responses to their comments, and we believe these clarifications also resolved most of their concerns. We have also updated our revised manuscript with new clarification and experiments.

We summarize the discussion and key points below for each reviewer.

---

## Global Responses

**Removal performance on RingID and WIND (xCat,E7z2)**: Addressed with reasoning and discussion. Also included a baseline comparison with Muller et al. to show that, at similar PSNR values, their attack is also not successful in removing RingID and WIND in **Table 4**.

**Baseline performance of Yang et al. and Zhao et al. (PQqk,E7z2)**: Addressed as it is due to their assumptions and limitations, respectively. Included baseline comparisons with Umarker and WAVES in **Table 2**.

---

## Other individual concerns

### Reviewer xCat

**Clarification on motivation using linear SVM**: Updated the paper with clarification.

**Significantly different VAE result analysis**: Clarified that indeed a model owner should consider building a significantly different VAE for better defense.

**Intuition behind mean pixel image**: Provided clarification.

### Reviewer rFDg

**Experiments limited to SDv1.4 and SDv2.0**: We have included new experiments on attacking FLUX.1-dev using the VAE from SDv1.4 (**Tables 3 and 4**), in addition to the existing comparison using the VAE from FLUX.1-dev to attack SDv1.4 (**Table 8**).

**Baseline comparison with UnMarker, VAEAttack (WAVES), Diffusion Attack**: Included baseline comparisons with UnMarker and VAEAttack (WAVES) in **Table 2**, and clarified that Diffusion Attack makes more assumptions than our setting.

**Generalization beyond latent-diffusion watermarks**: Clarified that the purpose of the paper is only to expose a key vulnerability in latent-diffusion watermarks.

### Reviewer E7z2

**RingID should embed the same signal as Tree-Ring**: This is factually incorrect, and we have provided a clarification.

**Requires access to VAE**: Clarified our attack is successful even with a significantly different VAE, and we only study latent-noise watermarks, where VAE is used in the detection pipeline.

**Single vs multi-watermark key**: Clarified that we use multi-watermark keys.

**Are perturbations epsilon ball bounded**: Clarified that we have ablation studies with epsilon ball bounded PGD attacks in Appendix B, where we show the current setting works better.

**Mathematical definitions of $l_2$ and $l_{\infty}$**: Provided definitions and clarified data ranges.

### Reviewer PQqk

**Too much in Appendix**: Updated paper to move some experiments from Appendix to the main paper.

**LPIPS vs PSNR**: Included a discussion in **Appendix G** to show that PSNR is more vital for watermark forgery/removal analysis as it accurately measures exact differences in image content. Added additional visual examples in **Appendix I**.

**Description of Lukas et al.'s work**: Clarified that we make a distinction between access to a VAE vs the entire diffusion model.

**Writing suggestions and clarifications**: Updated the paper, incorporating suggestions.

Thank you again for your time, effort, and service to the community.

Best regards,

Authors

---

### Meta-Review · Area_Chair_Dyqd · 2026-01-05

**Summary:**

Reviewers agree the paper makes a timely and potentially important observation about a many-to-one mapping induced by empty-prompt DDIM inversion, and leverages it to craft a practical black-box forgery/removal attack that often achieves strong forgery success with modest distortion.

However, multiple reviewers questioned the breadth of the claims: the evaluation was initially narrow, baseline coverage and protocol fairness needed strengthening, and the threat model/assumptions (latent-noise watermark specificity and access to a proxy VAE) required clearer positioning. A key technical concern is that removal appears substantially weaker than forgery, with failure cases on stronger schemes (e.g., RingID/WIND) limiting the claim of a generally effective removal method.

Overall, while the paper makes progress and the authors addressed several points during the discussion, the remaining gaps in generality, evaluation completeness, and the strength of the removal results fall short of the high technical and empirical bar expected at ICLR, leading to a recommendation to reject.

**Reviewer Concerns:**

The rebuttal addressed several clarity issues, expanded experiments (including additional baselines and proxy VAEs), and improved presentation, which partially satisfied reviewers xCat, rFDg, and PQqk. However, core concerns remain outstanding, including limited effectiveness over schemes (e.g., RingID, WIND), reliance on latent-noise-specific assumptions, and unresolved disagreement over evaluation metrics and claims of generality. These remaining issues weaken the paper’s overall impact and robustness.

**Reviewer Scores:**

Reviewer xCat would likely maintain a borderline accept score. Reviewer rFDg already increased their score slightly but would likely remain near the acceptance threshold. Reviewers E7z2 and PQqk would likely maintain reject or borderline scores.

---

### Decision · Program_Chairs · 2026-01-26

Reject